# Multi-Agent Reinforcement Learning for Active Voltage Control on Power Distribution Networks

**Jianhong Wang**[*]
Imperial College London
jianhong.wang16@imperial.ac.uk

**Wangkun Xu**[*]
Imperial College London
wangkun.xu18@imperial.ac.uk

**Yunjie Gu**[†]
University of Bath
yg934@bath.ac.uk

**Wenbin Song**
Shanghaitech University
songwb@shanghaitech.edu.cn

**Tim C. Green**[‡]
Imperial College London
t.green@imperial.ac.uk

## Abstract

This paper presents a problem in power networks that creates an exciting and yet challenging real-world scenario for application of multi-agent reinforcement learning (MARL). The emerging trend of decarbonisation is placing excessive stress on power distribution networks. Active voltage control is seen as a promising solution to relieve power congestion and improve voltage quality without extra hardware investment, taking advantage of the controllable apparatuses in the network, such as roof-top photovoltaics (PVs) and static var compensators (SVCs). These controllable apparatuses appear in a vast number and are distributed in a wide geographic area, making MARL a natural candidate. This paper formulates the active voltage control problem in the framework of Dec-POMDP and establishes an open-source environment. It aims to bridge the gap between the power community and the MARL community and be a drive force towards real-world applications of MARL algorithms. Finally, we analyse the special characteristics of the active voltage control problems that cause challenges (e.g. interpretability) for state-of-the-art MARL approaches, and summarise the potential directions.

## 1 Introduction

Multi-agent reinforcement learning (MARL) has demonstrated impressive performances on games (e.g., Chess [1], StarCraft [2, 3], and etc.) and robotics [4]. There are now increasing interests and thrusts to apply MARL in real-world problems. Power network is a natural test field for MARL algorithms. There are many problems in power networks involving the collaborative or competitive actions of a vast number of agents, such as market bidding [5], voltage control, frequency control, and emergency handling [6]. One of the obstacles for applying MARL in power networks is the lack of transparency and guarantee, which is unacceptable considering the importance of reliable power supply for the whole society. Therefore, it is sensible to start with problems that are less sensitive to reliability but hard to be solved by conventional methods.

Active voltage control in power distribution networks is such a candidate problem. The voltage control problem has been studied for years but only comes under spot light recently due to the increasing penetration of distributed resources, e.g. roof-top photovoltaics (PVs) [7]. The excessive active power injection may cause voltage fluctuations beyond the threshold of grid standards [8]. The voltage fluctuations can be relieved by exploiting the control flexibility of PV inverters themselves

---

[*]Equal contributions. † Correspondence to Yunjie Gu who is also an honorary lecturer at Imperial College London. ‡ Tim C. Green is also the co-director of the Energy Futures Laboratory (EFL).

35th Conference on Neural Information Processing Systems (NeurIPS 2021).

along with other controllable apparatuses, such as static var compensators (SVCs) and on load tap changers (OLTCs). An elaborate scheme is needed to coordinate these multiple apparatuses at scale to regulate voltage throughout the network with limited local information, which is called active voltage control [9–12]. The active voltage control problem has many interesting properties. (1) It is a combination of local and global problem, i.e., the voltage of each node is influenced by the powers (real and reactive) of all other nodes but the impact recedes with increasing distance between nodes. (2) It is a constrained optimisation problem where the constraint is the voltage threshold and the objective is the total power loss, but there is no explicit relationship between the constraint and the control action. (3) A distribution network has a radial topology involving a rich structure that can be taken as prior knowledge for control, but the node-branch parameters of the topology may be significantly uncertain. (4) Voltage control has a relatively large tolerance and less severe consequences if the control fails to meet standard requirements.

There have been several attempts to apply MARL in active voltage control [6, 13–16]. Each work on a particular case showed promising performances of state-of-the-art MARL approaches with modifications adapting to the active voltage control problem. However, it is not clear if these methods scale well to a larger network, and the robustness against different scenarios, such as penetration levels and load profiles, are yet to be investigated. There is no commonly accepted benchmark to provide the basis for fair comparison of different solutions.

To facilitate further research on this topic, and to bridge the gap between the power community and MARL community, we present a comprehensive test-bench and open-source environment for MARL based active voltage control. We formally define the active voltage control problem as a Dec-POMDP [17] that is widely acknowledged in the MARL community. We present an environment in Python and construct 3 scenarios with real public data that span from small scale (i.e. 6 agents) to large scale (i.e. 38 agents).

The contributions of this paper are summarised as follows: (1) We formally define the active voltage control problem as a Dec-POMDP and develop an open-source environment.[2] (2) We conduct large scale experimentation with 7 state-of-the-art MARL algorithms on different scenarios of the active voltage control problem. (3) We convert voltage constraints to barrier functions and observe the importance of designing an appropriate voltage barrier function from the experimental results. (4) By analysing the experimental results, we imply the possible challenges (e.g. interpretability) for MARL to solve the active voltage control problem and suggest potential directions for future works.

## 2   Related Work

**Traditional Methods for Active Voltage Control.**     Voltage rising and fluctuation problem in distribution networks has been studied for 20 years [18]. The traditional voltage regulation devices such as OLTC and capacitor banks [10] are often installed at substations and therefore may not be effective in regulating voltages at the far end of the line [19]. The emergence of distributed generation, such as root-top PVs, introduces new approaches for voltage regulation by the active reactive power control of grid-connected inverters [11]. The state-of-the-art active voltage control strategies can be roughly classified into two categories: (1) reactive power dispatch based on optimal power flow (OPF) [20, 21]; and (2) droop control based on local voltage and power measurements [22, 23]. Specifically, centralised OPF [24–26] minimises the power loss while fulfilling voltage constraints (e.g. power flow equation defined in Eq.1); distributed OPF [27, 28] used distributed optimization techniques, such as alternating direction method of multipliers (ADMM), to replace the centralised solver. The primary limitation of OPF is the need of exact system model [29]. Besides, solving constrained optimisation problem is time-consuming, so it is difficult to react to the rapid change of load profile [30]. On the other hand, droop control only depends on its local measurements, but its performance relies on the manually-designed parameters and is often sub-optimal due to the lack of global information [30]. It is possible to enhance droop control by distributed algorithms, but extra communications are needed [31, 19]. In this paper we investigate the possibility of applying MARL techniques on the active voltage control problem. Compared with the previous works on traditional methods, (1) MARL is model-free, so no exact system model is needed; and (2) the response of MARL is fast so as to handle rapid changes of environments (e.g. the intermittency of renewable energy).

---

[2] https://github.com/Future-Power-Networks/MAPDN.

**Multi-Agent Reinforcement Learning for Active Voltage Control.** In this section, we discuss the works that applied MARL to active voltage control problem in the power system community. [14, 16] applied MADDPG with the reactive power of inverters or static var compensators (SVCs) as control actions. [32] applied MADDPG with a manually designed voltage inner loop, so that agents set reference voltage instead of reactive power as their control actions. [13] applied MATD3 also with reactive power as control actions. [15] applied MASAC, where both reactive power and the curtailment of active power are used as control actions. In the above works, distribution networks are divided into regions, with each region controlled by a single agent [33]. It is not clear if these MARL approaches scales well for increasing number of agents. In particular, it is not clear if each single inverter in a distribution network can behave as an independent agent. In this work, we model the active voltage control problem as a Dec-POMDP [17], where each inverter is controlled by an agent. We propose Bowl-shape as a barrier function to represent voltage constraint as part of the reward. We build up an open-source environment for this specific problem that can be easily deployed with the existing MARL algorithms.

**Concurrent Works.** L2RPN [34] is an environment mainly for the centralised control over power transmission networks with node-splitting and lines switch-off for topology management. Gym-ANM [35] is an environment for centralised power distribution network management. Compared with these 2 works, our environment mainly focuses on solving the active voltage control problem in power distribution networks, with the decentralised/distributed manner. Moreover, we provide more complicated scenarios with the real-world data. Finally, our environment can be easily and flexibly extended with more network topologies and data, thanks to the supports of PandaPower [36] and SimBench [37].

# 3 Background

## 3.1 Power Distribution Network

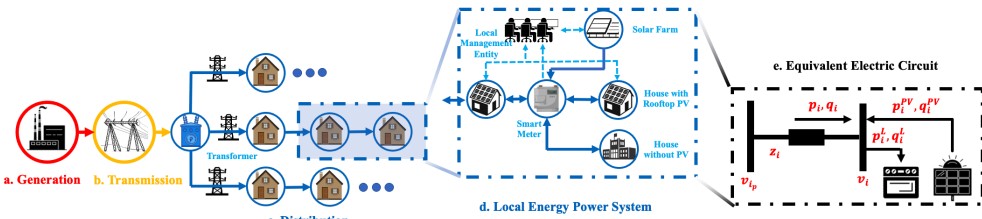

Figure 1: Illustration on distribution network (block a-b-c) under PV penetration. The solid and dotted lines represent the power and information flows respectively. Block d is the detailed version of distribution network and block e is the circuit model of block d.

An electric power distribution network is illustrated in Figure 1 stage a to c. The electricity is generated from power plant and transmitted through transmission lines. Muti-stage transformers are applied to reduce the voltage levels while the electricity is being delivered to the distribution network. The electricity is then consumed by residential and industrial clients. A typical PV unit consists of PV panels and voltage-source inverters which can be installed either on roof-top or in the solar farm. Conventionally, there exist management entities such as distributed system operator (DSO) monitoring and operating the PV resources through the local communication channels. With emergent PV penetration, distribution network gradually grows to be an active participant in power networks that can deliver power and service to its users and the main grid (see the bidirectional power flows in Figure 1 stage-d).

**System Model and Voltage Deviation.** In this paper, we consider medium (10-24kV) and low (0.23-1kV) voltage distribution networks where PVs are highly penetrated. We model the distribution network in Figure 1 as a tree graph $\mathcal{G} = (V, E)$, where $V = \{0, 1, \ldots, N\}$ and $E = \{1, 2, \ldots, N\}$ represent the set of nodes (buses) and edges (branches) respectively [24]. Bus 0 is considered as the connection to the main grid, balancing the active and reactive power in the distribution network. For each bus $i \in V$, let $v_i$ and $\theta_i$ be the magnitude and phase angle of the complex voltage and $s_j = p_i + jq_i$ be the complex power injection. Then the active and reactive power injection can be

defined as follows:

$$p_i^{PV} - p_i^L = v_i^2 \sum_{j \in V_i} g_{ij} - v_i \sum_{j \in V_i} v_j \left( g_{ij} \cos \theta_{ij} + b_{ij} \sin \theta_{ij} \right), \quad \forall i \in V \setminus \{0\}$$

$$q_i^{PV} - q_i^L = -v_i^2 \sum_{j \in V_i} b_{ij} + v_i \sum_{j \in V_i} v_j \left( g_{ij} \sin \theta_{ij} + b_{ij} \cos \theta_{ij} \right), \quad \forall i \in V \setminus \{0\} \tag{1}$$

where $V_i := \{j \mid (i,j) \in E\}$ is the index set of buses connected to bus $i$. $g_{ij}$ and $b_{ij}$ are the conductance and susceptance on branch $(i,j)$. $\theta_{ij} = \theta_i - \theta_j$ is the phase difference between bus $i$ and $j$. $p_i^{PV}$ and $q_i^{PV}$ are the active power and reactive power of the PV on the bus $i$ (that are zeros if there is no PV on the bus $i$). $p_i^L$ and $q_i^L$ are the active power and reactive power of the loads on the bus $i$ (that are zeros if there is no loads on the bus $i$). Eq.1 can represent the power system dynamics which is essential for solving the power flow problem and active voltage control problem [38] (details in Appendix A.1-A.2). For safe and optimal operation, 5% voltage deviation is usually allowed, i.e., $v_0 = 1.0$ per unit ($p.u.$) and $0.95\ p.u. \leq v_i \leq 1.05\ p.u., \forall i \in V \setminus \{0\}$. When the load is heavy during the nighttime, the end-user voltage could be smaller than $0.95\ p.u.$ [39]. In contrast, to export its power, large penetration of $p_i^{PV}$ leads to reverse current flow that would increase $v_i$ out of the nominal range (Figure 1-d) [18, 40].

**Optimal Power Flow.**   In this paper, OPF is regarded as an optimization problem minimizing the total power loss subject to the power balance constraints defined in Eq.1, PV reactive power limits, and bus voltage limits [41]. As the centralized OPF has full access to the system topology, measurements, and PV resources, it provides the optimal active voltage control performance and can be used as a benchmark method (details in Appendix A.3). However, the performance of the OPF depends on the accuracy of the grid model and the optimisation is time-consuming which makes it difficult to be deployed online.

**Droop Control.**  To regulate local voltage deviation, the standard droop control defines a piece-wise linear relationship between PV reactive power generation and voltage deviation at a bus equipped with inverter-based PVs [30, 42] (details in Appendix A.4). It is a fully decentralised control and ignore both the total voltage divisions and the power loss.

## 3.2   Dec-POMDP

Multi-agent reinforcement learning (MARL) is an extension from the reinforcement learning problem, with multiple agents in the same environment. The problem of MARL for cooperation among agents is conventionally formulated as a Dec-POMDP [17]. It is usually formulated as a tuple such that $\langle \mathcal{I}, \mathcal{S}, \mathcal{A}, \mathcal{O}, \mathcal{T}, r, \Omega, \rho, \gamma \rangle$. $\mathcal{I}$ is an agent set; $\mathcal{S}$ is a state set; $\mathcal{A} = \times_{i \in \mathcal{I}} \mathcal{A}_i$ is the joint action set, where $\mathcal{A}_i$ is each agent's action set; $\mathcal{O} = \times_{i \in \mathcal{I}} \mathcal{O}_i$ is the joint observation set, where $\mathcal{O}_i$ is each agent's observation set; $\mathcal{T} : \mathcal{S} \times \mathcal{A} \times \mathcal{S} \to [0,1]$ is a transition probability function that describes the dynamics of an environment; $r : \mathcal{S} \times \mathcal{A} \to \mathbb{R}$ is a global reward function that describes the award to the whole agents given their decisions; $\Omega : \mathcal{S} \times \mathcal{A} \times \mathcal{O} \to [0,1]$ describes the perturbation of the observers (or sensors) for agents' joint observations over the states after decisions; $\rho : \mathcal{S} \to [0,1]$ is a probability function of initial states; and $\gamma \in (0,1)$ is a discount factor. The objective of Dec-POMDP is finding an optimal joint policy $\pi = \times_{i \in \mathcal{I}} \pi_i$ that solves $\max_\pi \mathbb{E}_\pi \left[ \sum_{t=0}^{\infty} \gamma^t r_t \right]$.

# 4   Distributed Active Voltage Control Problem

## 4.1   Problem Formulation

For the ease of operations, a large-scale power network is divided into multiple regions and there are several PVs installed into each region that is managed by the responsible distribution network owner. Each PV is with an inverter that generates reactive power to control the voltage around a stationary value denoted as $v_{\text{ref}}$. In our problem, we assume that the PV ownership (e.g. individual homeowners or enterprise) is independent and separated from the distribution network ownership (DNO) [33]. Specifically, a PV owner is responsible for operating a PV infrastructure. In other words, each PV is able to be controlled under a distributed manner so that it is natural to be considered as an agent. To enforce the safety of distribution power networks, all agents (i.e. PVs) within a region share the

observation of this region.[3] Since each agent can only observe partial information of the whole gird and maintaining the safety of the power network is a common goal among agents, it is reasonable to model the problem as a Dec-POMDP [17] that can be mathematically described as a 10-tuple such that $\langle \mathcal{I}, \mathcal{S}, \mathcal{A}, \mathcal{R}, \mathcal{O}, \mathcal{T}, r, \Omega, \rho, \gamma \rangle$, where $\rho$ is the probability distribution for drawing the initial state and $\gamma$ is the discount factor.

**Agent Set.** There is a set of agents controlling a set of PV inverters denoted as $\mathcal{I}$. Each agent is located at some node in $\mathcal{G}$ (i.e. a graph representing the power network defined as before). We define a function $g : \mathcal{I} \to V$ to indicate the node where an agent is located.

**Region Set.** The whole power network is separated into $M$ regions, whose union is denoted as $\mathcal{R} = \{\mathcal{R}_k \subset V \mid k < M, k \in \mathbb{N}\}$, where $\bigcup_{\mathcal{R}_k \in \mathcal{R}} \mathcal{R}_k \subseteq V$ and $\mathcal{R}_{k_1} \cap \mathcal{R}_{k_2} = \emptyset$ if $k_1 \neq k_2$. We define a function $f : V \to \mathcal{R}$ that maps a node to the region where it is involved.

**State and Observation Set.** The state set is defined as $\mathcal{S} = \mathcal{L} \times \mathcal{P} \times \mathcal{Q} \times \mathcal{V}$, where $\mathcal{L} = \{(\mathbf{p}^L, \mathbf{q}^L) : \mathbf{p}^L, \mathbf{q}^L \in (0, \infty)^{|V|}\}$ is a set of (active and reactive) powers of loads; $\mathcal{P} = \{\mathbf{p}^{PV} : \mathbf{p}^{PV} \in (0, \infty)^{|\mathcal{I}|}\}$ is a set of active powers generated by PVs; $\mathcal{Q} = \{\mathbf{q}^{PV} : \mathbf{q}^{PV} \in (0, \infty)^{|\mathcal{I}|}\}$ is a set of reactive powers generated by PV inverters at the preceding step; $\mathcal{V} = \{(\mathbf{v}, \theta) : \mathbf{v} \in (0, \infty)^{|V|}, \theta \in [-\pi, \pi]^{|V|}\}$ is a set of voltage wherein $\mathbf{v}$ is a vector of voltage magnitudes and $\theta$ is a vector of voltage phases measured in radius. $v_i, p_i^L, q_i^L, p_i^{PV}$ and $q_i^{PV}$ are denoted as the components of the vectors $\mathbf{v}, \mathbf{p}^L, \mathbf{q}^L, \mathbf{p}^{PV}$ and $\mathbf{q}^{PV}$ respectively. We define a function $h : \mathbb{P}(V) \to \mathbb{P}(\mathcal{S})$ that maps a subset of $V$ to its correlated measures, where $\mathbb{P}(\cdot)$ denotes the power set. The observation set is defined as $\mathcal{O} = \times_{i \in \mathcal{I}} \mathcal{O}_i$, where $\mathcal{O}_i = (h \circ f \circ g)(i)$ indicates the measures within the region where agent $i$ is located.

**Action Set.** Each agent $i \in \mathcal{I}$ is equipped with a continuous action set $\mathcal{A}_i = \{a_i : -c \leq a_i \leq c, c > 0\}$. The continuous action represents the ratio of maximum reactive power it generates, i.e., the reactive power generated from the $k$th PV inverter is $q_k^{PV} = a_k \sqrt{(s_k^{\max})^2 - (p_k^{PV})^2}$, where $s_k^{\max}$ is the maximum apparent power of the $k$th node that is dependent on the physical capacity of the PV inverter.[4,5] If $a_k > 0$, it means penetrating reactive powers to the distribution network. If $a_k < 0$, it means absorbing reactive powers from the distribution network. The value of $c$ is usually selected as per the loading capacity of a distribution network, which is for the safety of operations. The joint action set is denoted as $\mathcal{A} = \times_{i \in \mathcal{I}} \mathcal{A}_i$.

**State Transition Probability Function.** Since the state includes the last action and the change of loads is random (that theoretically can be modelled as any probabilistic distribution), we can naturally define the state transition probability function as $\mathcal{T} : \mathcal{S} \times \mathcal{A} \times \mathcal{S} \to [0, 1]$ that follows Markov decision process. Specifically, $\mathcal{T}(\mathbf{s}_{t+1} | \mathbf{s}_t, \mathbf{a}_t) = Pr(\mathbf{s}_{t+1} | \delta(\mathbf{s}_t, \mathbf{a}_t))$, where $\mathbf{a}_t \in \mathcal{A}$ and $\mathbf{s}_t, \mathbf{s}_{t+1} \in \mathcal{S}$. $\delta(\mathbf{s}_t, \mathbf{a}) \mapsto \mathbf{s}_{t+\tau}$ denotes the solution of the power flow, whereas $Pr(\mathbf{s}_{t+1} | \mathbf{s}_{t+\tau})$ describes the change of loads (i.e. highly correlated to the user behaviours). $\tau \ll \Delta t$ is an extremely short interval much less than the time interval between two controls (i.e. a time step) and $\Delta t = 1$ in this paper.

**Observation Probability Function.** We now define the observation probability function. In the context of electric power network, it describes the measurement errors that may occur in sensors. Mathematically, we can define it as $\Omega : \mathcal{S} \times \mathcal{A} \times \mathcal{O} \to [0, 1]$. Specifically, $\Omega(\mathbf{o}_{t+1} | \mathbf{s}_{t+1}, \mathbf{a}_t) = \mathbf{s}_{t+1} + \mathcal{N}(\mathbf{0}, \Sigma)$, where $\mathcal{N}(\mathbf{0}, \Sigma)$ is an isotropic multi-variable Gaussian distribution and $\Sigma$ is dependent on the physical properties of sensors (e.g. smart meters).

**Reward Function.** The reward function is defined as follows:

$$r = -\frac{1}{|V|} \sum_{i \in V} l_v(v_i) - \alpha \cdot l_q(\mathbf{q}^{PV}), \tag{2}$$

where $l_v(\cdot)$ is a voltage barrier function and $l_q(\mathbf{q}^{PV}) = \frac{1}{|\mathcal{I}|} ||\mathbf{q}^{PV}||_1$ is the reactive power generation loss (i.e. a type of power loss approximation easy for computation). The objective is to control the voltage within a safety range around $v_{\text{ref}}$, while the reactive power generation is as less as possible, i.e.,

---

[3] Sharing observation in this problem is reasonable, since only the sensor measurements (e.g. voltage, active power, etc.) are shared, which are not directly related to the commercial profits [33]. The observation of each PV is collected by the distribution network owner and then the full information within the region is sent to each agent.

[4] Note that the reactive power range actually dynamically changes at each time step.

[5] Yielding $(q_k^{PV})_t$ at each time step $t$ is equivalent to yielding $\Delta_t(q_k^{PV})$ (i.e. the change of reactive power generation at each time step), since $(q_k^{PV})_t = (q_k^{PV})_{t-1} + \Delta_t(q_k^{PV})$. For easily satisfying the safety condition, we directly yield $q_k^{PV}$ at each time step in this work.

$l_q(\mathbf{q}^{PV}) < \epsilon$ and $\epsilon > 0$. Similar to the mathematical tricks used in $\beta$-VAE [43], by KKT conditions we can transform a constrained reward to a unconstrained reward with a Lagrangian multiplier $\alpha \in (0, 1)$ shown in Eq.2. Since $l_v(\cdot)$ is not easy to define in practice (i.e., it affects $l_q(\mathbf{q}^{PV})$), we aim to study for a good choice in this paper.

**Objective Function.** The objective function of this problem is $\max_\pi \mathbb{E}_\pi[\sum_{t=0}^{\infty} \gamma^t r_t]$, where $\pi = \times_{i \in \mathcal{I}} \pi_i$; $\pi_i : \bar{\mathcal{O}}_i \times \mathcal{A}_i \to [0, 1]$ and $\bar{\mathcal{O}}_i = (\mathcal{O}_i^\tau)_{\tau=1}^h$ is a history of observations with the length as $h$. Literally, we need to find an optimal joint policy $\pi$ to maximize the discounted cumulative rewards.

## 4.2 Voltage Barrier Function

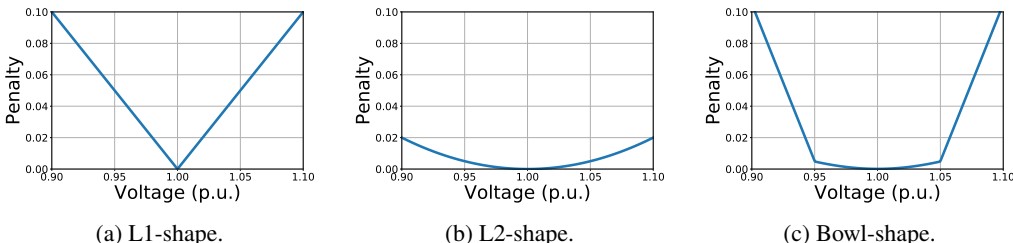

(a) L1-shape.       (b) L2-shape.       (c) Bowl-shape.

Figure 2: This figure shows 3 voltage barrier functions, where L1-shape and L2-shape are 2 baselines while Bowl-shape is proposed in this paper.

We define $v_{\text{ref}} = 1\ p.u.$ in this paper, and the voltage needs to be controlled within the safety range from $0.95\ p.u.$ to $1.05\ p.u.$, which sets the constraint of control. The voltage constraint is difficult to be handled in MARL, so we use a barrier function to represent the constraint. L1-shape (see Figure 2a) was most frequently used in the previous work [13–15], however, this may lead to wasteful reactive power generations since $\frac{|\Delta l_v|}{\alpha |\Delta l_q|} \gg 1$ within the safety range of voltage. Although L2-shape (see Figure 2b) may alleviate this problem, it may be slow to guide the policy outside the safety range. To address these problems, we propose a barrier function called Bowl-shape that combines the advantages of L1-shape and L2-shape. It gives a steep gradient outside the safety range, while it provides a slighter gradient as voltage tends to the $v_{\text{ref}}$ that enables $\frac{|\Delta l_v|}{\alpha |\Delta l_q|} \to 0$ as $v \to v_{\text{ref}}$.

# 5 Experiments

## 5.1 Experimental Settings

**Power Network Topology.** Two MV networks, IEEE 33-bus [44] and 141-bus [45] are modified as systems under test.[6] To show the flexibility on network with multi-voltage levels, we construct a 110kV-20kV-0.4kV (high-medium-low voltage) 322-bus network using benchmark topology from SimBench [37]. For each network, a main branch is firstly determined and the control regions are partitioned by the shortest path between the terminal bus and the coupling point on the main branch. Each region consists of 1-4 PVs dependent on various regional sizes. The specific network descriptions and partitions are shown in Appendix D.1. To give a picture of the tasks, we demonstrate the 33-bus network in Figure 3.

**Data Descriptions.** The load profile of each network is modified based on the real-time Portuguese electricity consumption accounting for 232 consumers of 3 years.[7] To highlight the differences between residential and industrial users, we randomly perturb $\pm 5\%$ on the default power factors defined in the case files and accordingly generate real-time reactive power consumption. The solar data is collected from Elia group,[8] i.e. a Belgiums power network operator. The load and PV data are then interpolated with 3-min resolution that is consistent with the real-time control period in the grid. To distinguish among different solar radiation levels in various regions, the 3-year PV generations

---

[6]The original topologies and parameters can be found in MATPOWER [46] description file such as `https://github.com/MATPOWER/matpower/tree/master/data`.

[7]`https://archive.ics.uci.edu/ml/datasets/ElectricityLoadDiagrams20112014`.

[8]`https://www.elia.be/en/grid-data/power-generation/solar-pv-power-generation-data`.

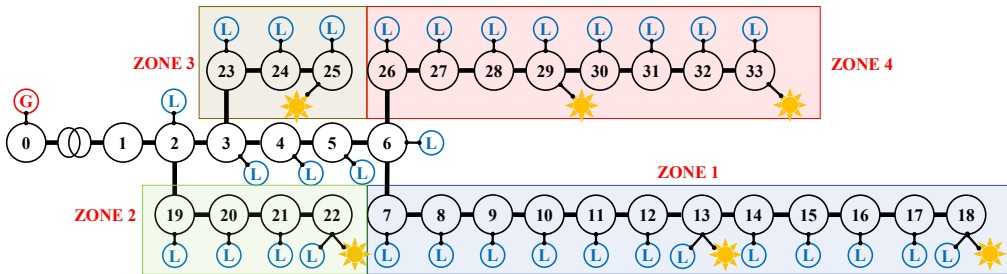

Figure 3: Illustration on 33-bus network. Each bus is indexed by a circle with a number. 4 control regions are partitioned by the smallest path from the terminal to the main branch (bus 1-6). We control the voltages on bus 2-33 whereas bus 0-1 represent the substation or main grid with the constant voltage and infinite active and reactive power capacity. **G** represents an external generator; small **L**s represent loads; and the sun emoji represents the location where a PV is installed.

from 10 cites/regions are collected and PVs in the same control region possess the same generation profiles. We define the PV penetration rate ($PR$) as the ratio between rated PV generation and rated load consumption. In this paper, we set $PR \in \{2.5, 4, 2.5\}$ as the default $PR$ for different topologies. We oversize each PV inverter by 20% of its maximum active power generation to satisfy the IEEE grid code [42]. Besides, each PV inverter is considered to be able to generate reactive power in the STATCOM mode during night [47]. The median and 25%-75% quantile shadings of PV generations, and the mean and minima-maxima shading of the loads are illustrated in Figure 4. The details can be found in Appendix D.2.

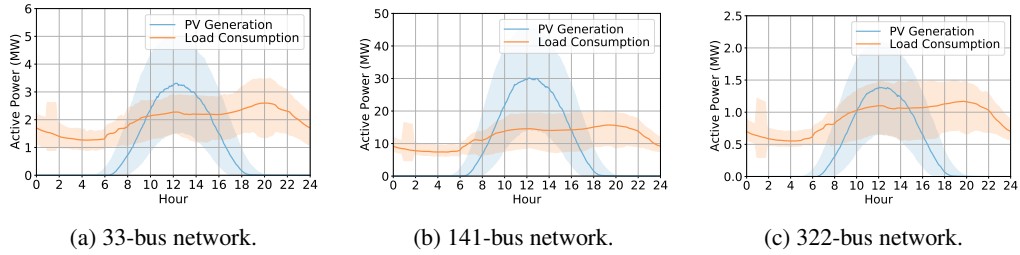

(a) 33-bus network.  (b) 141-bus network.  (c) 322-bus network.

Figure 4: Active PV generations and load consumption.

**MARL Simulation Settings.** We now describe the simulation settings standing by the view of MARL. In 33-bus network, there are 4 regions with 6 agents. In 141-bus network, there are 9 regions with 22 agents. In 322-bus network, there are 22 regions with 38 agents. The discount factor $\gamma$ is set to 0.99. $\alpha$ in Eq.2 is set to 0.1. To guarantee the safety of distribution networks, we manually set the range of actions for each scenario, with $[-0.8, 0.8]$ for 33-bus network, $[-0.6, 0.6]$ for 141-bus network, and $[-0.8, 0.8]$ for 322-bus network. During training, we randomly sample the initial state for an episode and each episode lasts for 240 time steps (i.e. a half day). Every experiment is run with 5 random seeds and the test results during training are given by the median and the 25%-75% quantile shading. Each test is conducted every 20 episodes with 10 randomly selected episodes for evaluation.

**Evaluation Metrics.** In experiments, we use two metrics to evaluate the performance of algorithms.

- *Controllable rate (CR)*: It calculates the ratio of time steps where all buses' voltages being under control during each episode.

- *Power loss (PL)*: It calculates the average of the total power loss over all buses per time step during each episode.

We aim to find algorithms and reward functions with high CR and low PL.

**MARL Algorithm Settings.** We evaluate the performances of state-of-the-art MARL algorithms, i.e. IDDPG [48], MADDPG [49], COMA [50], IPPO [2], MAPPO [3], SQDDPG [48] and MATD3 [51] on this real-world problem with continuous actions. Since the original COMA can only work

for discrete actions, we conduct some modifications so that it can work for continuous actions (see Appendix B). The details of settings are shown in Appendix C.

## 5.2 Main Results

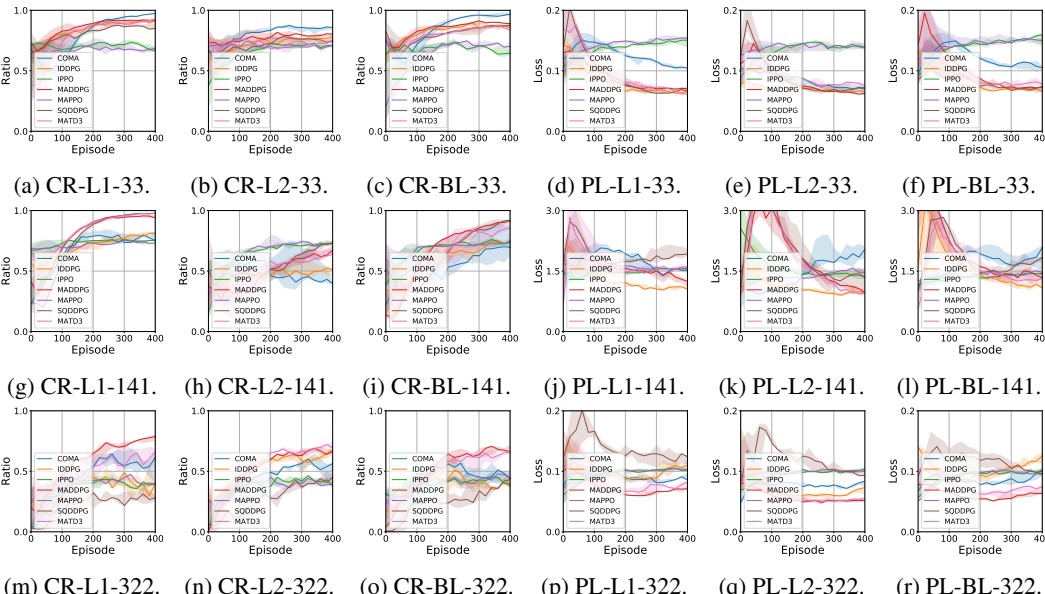

| (a) CR-L1-33. | (b) CR-L2-33. | (c) CR-BL-33. | (d) PL-L1-33. | (e) PL-L2-33. | (f) PL-BL-33. |
|---|---|---|---|---|---|
| (g) CR-L1-141. | (h) CR-L2-141. | (i) CR-BL-141. | (j) PL-L1-141. | (k) PL-L2-141. | (l) PL-BL-141. |
| (m) CR-L1-322. | (n) CR-L2-322. | (o) CR-BL-322. | (p) PL-L1-322. | (q) PL-L2-322. | (r) PL-BL-322. |

Figure 5: Median CR and PL of algorithms with different voltage barrier functions. The sub-caption indicates metric-Barrier-scenario and BL is the contraction of Bowl.

**Algorithm Performances.** We first show the main results of all algorithms on all scenarios in Figure 5. MADDPG and MATD3 generally perform well on all scenarios with different voltage barier functions. COMA performs well over CR on 33-bus networks and the performances fall on the large scale scenarios, but its PL is high. Similarly, SQDDPG performs generally well over CR on 33-bus and 141-bus networks, but it performs the worst on 322-bus networks and its PL on 141-bus networks is high. This reveals the limitations of COMA and SQDDPG on the scaling to many agents. Although MAPPO and IPPO performed well on games [2, 3], their performances on the real-world power network problems are poor. The most probable reason could be that the variations of dynamics and uncertainties in distribution networks are far faster and more complicated than games and their conservative policy updates cannot fast respond. IDDPG performs at the middle place in all scenarios, which may be due to non-stationary dynamics caused by multiple agents [49].

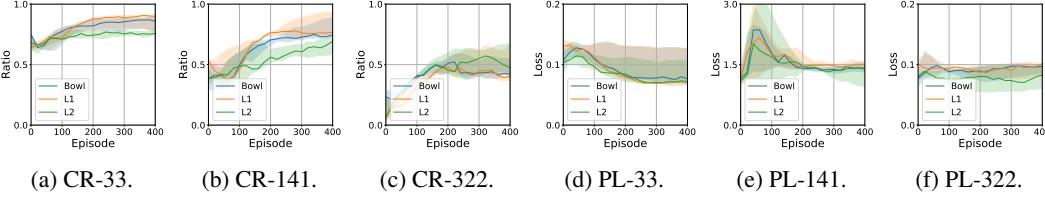

| (a) CR-33. | (b) CR-141. | (c) CR-322. | (d) PL-33. | (e) PL-141. | (f) PL-322. |
|---|---|---|---|---|---|

Figure 6: Median performance of overall algorithms with different voltage barrier functions. The sub-caption indicates metric-scenario.

**Voltage Barrier Function Comparisons.** To study the effects of different voltage barrier functions, we show the median performance of overall 7 MARL algorithms in Figure 6. It can be observed that Bowl-shape can preserve the high CR, while maintain the low PL on 33-bus and 141-bus networks. Although L1-shape can achieve the best CR on 33-bus and 141-bus networks, its PL on the 141-bus network is the highest. L2-shape performs the worst on 33-bus and 141-bus networks, but performs the best on the 322-bus network with the highest CR and the lowest PL. The reason could be that its slighter gradients is more suitable for the adaption among many agents. From the results, we can

conclude that L1-shape, Bowl-shape and L2-shape are the best choices for the 33-bus network, the 141-bus network and the 322-bus network respectively.

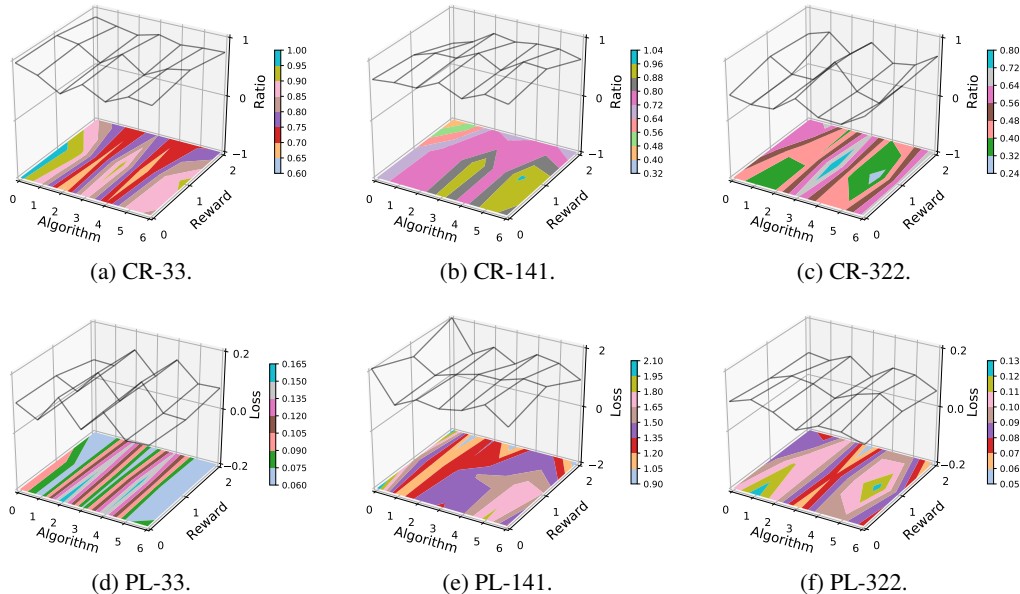

| (a) CR-33. | (b) CR-141. | (c) CR-322. |

| (d) PL-33. | (e) PL-141. | (f) PL-322. |

Figure 7: Median performances of overall algorithms trained with various rewards consist of distinct voltage barrier functions shown in 3D surfaces. The sub-caption indicates metric-scenario.

**Diverse Algorithm Performances under Distinct Rewards.** To clearly show the relationship between algorithms and reward functions, we also plot 3D surfaces over CR and PL w.r.t. algorithm types and reward types (consisting of distinct voltage barrier functions) in Figure 7. It is obvious that the performances of algorithms are highly correlated with the reward types. In other words, the same algorithm could perform diversely even trained by different reward functions with the same objective but different shapes.

## 5.3 Comparison between MARL and Traditional Control Methods

To compare MARL algorithms with the traditional control methods, we conduct a series of tests on various network topologies (i.e. 33-bus, 141-bus, and 322-bus networks). MADDPG trained by Bowl-shape is selected as the candidate for MARL. The traditional control methods that we select are OPF [24] and droop control [22]. For conciseness, we only demonstrate the voltages and powers on a typical bus with a PV installed (i.e. one of the most difficult buses to control) during a day (i.e. 480 consecutive time steps) in summer and winter respectively. The results for the 33-bus network is presented here and the results for other typologies are given in the Appendix E.2. From Figure 8, it can be seen that all methods control the voltage within the safety range in both summer and winter. Additionally, the power losses of MARL are lower than droop control but higher than OPF. This phenomenon is possibly due to the fact that droop control is a fully distributed algorithm which cannot explicitly reduce the power loss and OPF is a centralised algorithm with the known system model, while MARL lies between these 2 types of algorithms. This implies that MARL may outperform droop control for particular cases but it needs to be kept in mind that the droop gain used here may not be optimal. It is also worth noting that the control actions of MARL has a similar feature to droop control in the 33-bus case, and this feature is also observed in the 141-bus and 322-bus cases (see Figure 14-15 in the Appendix E.2 for details). However, this droop-like behaviour is missing in the winter case of 322-bus network, where the MARL control action is opposite to the desired direction, resulting in higher power losses and voltage deviation. This might be due to the over-generation as the MARL share the same policy for different seasons, so that it leads to the problem of relative overgeneralisation [52]. In summary, MARL seems able to learn the features of droop control but fails to outperform droop control significantly, and is even worse than droop control for certain cases.

As a result, there is still significant headroom of improvement for MARL, in performance, robustness, and interpretability.

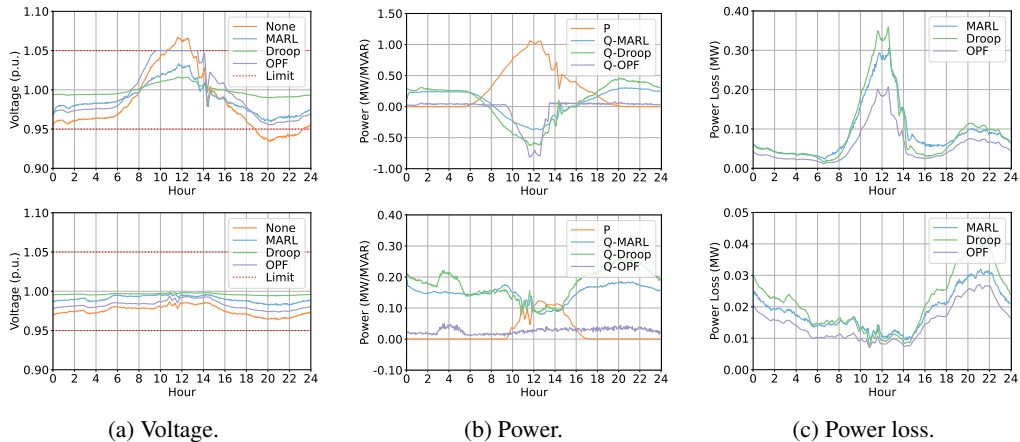

(a) Voltage.          (b) Power.          (c) Power loss.

Figure 8: Compare MARL with traditional control methods on bus 18 during a day for 33-bus network. 1st row: results for a summer day. 2nd row: results for a winter day. None and limit in (a) represent the voltage with no control and the safety voltage range respectively. P and Q in (b) indicate the PV active power and the reactive power by various methods.

## 5.4 Discussion

We now discuss the phenomenons that we observe from the experimental results.

- It is obvious that the voltage barrier function may impact the performance of an algorithm (even with tiny changes for the common goal). This may be of general importance as many real-world problems may contain constraints that are not indirect in the objective function. An overall methodology is needed for designing barrier functions in state-constraint MARL.

- MARL may scale well for the number of agents and the complexity of networks, and only requires a very low control rate for active voltage control.

- The results above show the evidence that MARL may behave diversely from the traditional control methods. Some of the learnt behaviours may induce better performance, but some others deteriorate the performance. This shows the promising benefits of MARL in industrial applications but also highlights the drawbacks in interpretability and robustness.

- The combination of learning algorithms with domain knowledge is a potential roadmap towards interpretable MARL. For the active voltage control problem, the domain knowledge may present as network topology, inner control loops (say droop control), and load pattern. The exploitation of such domain knowledge reduces the dimensions of MARL exploration space and may offer a lower bound of performance as a guarantee. Encoding domain knowledge such as the network topology as a priori for model-based MARL is also a potential direction.

## 6   Conclusion

This paper investigates the potential of applying multi-agent reinforcement learning (MARL) to the active voltage control in power distribution networks. We firstly formulate this problem as a Dec-POMDP and then study the behaviours of MARL with various voltage barrier functions (reflecting voltage constraints as barrier penalties). Moreover, we compare the behaviours of MARL with the traditional control methods (i.e. droop control and OPF), and observe that MARL is possible to generate inexplicable behaviours, so an interpretable and trustable algorithm is highly desired for industrial applications. Finally, the environment used in this work is open-sourced and easy to follow so that the machine learning community is able to contribute to this challenging real-world problem.

## Acknowledgement

This work is supported by the Engineering and Physical Sciences Research Council of UK (EPSRC) under awards EP/S000909/1.

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
