# Multi-Agent Reinforcement Learning for Active Voltage Control on Power Distribution Networks: Supplementary Material

## A  Additional Background of Voltage Control Problem

### A.1  Power Flow Problem

Recall the power balance equations:

$$
\begin{aligned}
p_i^{PV} - p_i^L &= v_i^2 \sum_{j \in V_i} g_{ij} - v_i \sum_{j \in V_i} v_j \left( g_{ij} \cos\theta_{ij} + b_{ij} \sin\theta_{ij} \right), \quad \forall i \in V \setminus \{0\} \\
q_i^{PV} - q_i^L &= -v_i^2 \sum_{j \in V_i} b_{ij} + v_i \sum_{j \in V_i} v_j \left( g_{ij} \sin\theta_{ij} + b_{ij} \cos\theta_{ij} \right), \quad \forall i \in V \setminus \{0\}
\end{aligned}
\tag{1}
$$

The power flow problem is designed to find the steady-state operation point of power system. After measuring power injections $p_i^{PV} - p_i^L$ and $q_i^{PV} - q_i^L$, the bus voltages $v_i \angle \theta_i$ can be retrieved by iteratively solving Eq.1 using Newton-Raphson or Gauss-Seidel method [1]. The power plow serves as the fundamental role in grid planning and security assessment by locating any voltage deviations. It is also used to generate the observations during MARL training.

### A.2  Voltage Deviation and Control

**Two-Bus Network.**  To intuitively show how voltage is varied by PVs and how PV inverters can participate in voltage control, we give an example for a two-bus distribution network in Figure 1. In Figure 1, $z_i = r_i + jx_i$ represents the impedance on branch $i$; $r_i$ and $x_i$ are resistance and reactance on branch $i$, respectively; $p_i^L$ and $q_i^L$ denote active and reactive power consumption, respectively; $p_i^{PV}$ and $q_i^{PV}$ indicate active and reactive PV power generation, respectively. The parent bus voltage $v_{i_p}$ is set as reference for the two-bus network.

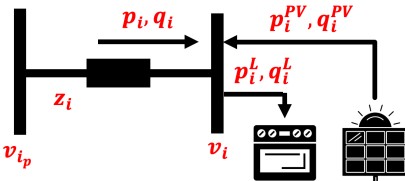

Figure 1: Two-bus electric circuit of the distribution network.

The voltage drop $\Delta v_i = v_{i_p} - v_i$ in Figure 1 can be approximated as follows:

$$
\Delta v_i = \frac{r_i(p_i^L - p_i^{PV}) + x_i(q_i^L - q_i^{PV})}{v_i}
\tag{2}
$$

The power loss of the 2-bus network in Figure 1 can be written as:

$$P_{\text{loss}} = \frac{(p_i^L - p_i^{PV})^2 + (q_i^L - q_i^{PV})^2}{v_{i_p}^2} \cdot r_i \tag{3}$$

**Traditional Voltage Control Methods.** Conventionally, PVs are not allowed to participate in voltage control so that $q_i^{PV}$ is restricted to 0 by the grid code. To export its power, large penetration of $p_i^{PV}$ may increase $v_i$ out of its safe range, causing reverse current flow [2, 3]. Voltage control devices, such as shunt capacitor (SC) and step voltage regulator (SVR) are usually equipped in the network to maintain the voltage level [4]. Nonetheless, these methods cannot respond to intermittent solar radiation, e.g. frequent voltage fluctuation due to cloud cover [5]. Additionally, with the rising PV penetration in the network, the operation of traditional regulators would be at their control limit (i.e. runaway condition) [6].

**Inverter-based Volt/Var Control.** To adapt to the continually rising PV penetration, grid-support services, such as voltage and reactive power control are required for every new-installed PV by the latest grid code IEEE Std-1547™-2018 [7]. For instance, the PV reactive power can be regulated by the PV-inverter under partial static synchronous compensator (STATCOM) mode [8]. Depending on the voltage deviation levels, the inverter can inject or absorb different amount of reactive power exceeding its capacity [9]. This control method is then named as Volt/Var control, as the reactive power (with unit VAR) is determined by the voltage (with unit Volt). Intuitively by Eq.2, when the voltage increases due to large PV penetration in the lunch-time, the PV inverter absorbs reactive power while during the night-time, the full inverter capacity is used to balance voltage fluctuation caused by increasing load [6].

Note that the only control variable in Eq.2 and Eq.3 is $q_i^{PV}$ which represents the reactive power generated by PV. Based on Eq.2, to enforce zero voltage deviation, the reactive power should satisfy the following condition

$$q_i^{PV} = \frac{r_i}{x_i}(p_i^L - p_i^{PV}) + q_i^L \tag{4}$$

Since the ratio $r_i/x_i$ in the distribution network is extremely large, $q_i^{PV}$ could become negative (i.e. absorbing reactive power) with great magnitude during the period of the peak PV injection (i.e., $p_i^{PV} \gg p_i^L$).

From Eq.3, to achieve the least power loss, $q_i^{PV}$ needs to be equal to $q_i^L$ (i.e. no reactive power injection). This result may conflict with the voltage control target in Eq.4, implying that it is hard to simultaneously maintain safe voltage levels and minimise the power losses, even for the two-bus network.

This section only demonstrates a 2-bus network which has linear relationship between voltage deviation and PV reactive power. Although the power systems in real world are non-linear and more complex, they are with the same phenomenon on the contradiction between voltage control and power loss minimisation.

## A.3  Optimal Power Flow

The optimal power flow (OPF) considered in this paper can be briefly formulated as:

$$
\begin{aligned}
\min_{q_i^{PV}} \quad & p_0 \\
\text{s.t.} \quad & \text{Eq.1} \\
& |q_i^{PV}| \le q_{i,\max}^{PV}, \quad i \in V^{PV} \\
& v_{i,\min} \le v_i \le v_{i,\max}, \quad i \in V \setminus 0 \\
& v_0 = v_{\text{ref}}
\end{aligned}
\tag{5}
$$

where $p_0$ and $v_0$ are the active power and reference voltage of the slack bus, respectively. $V^{PV}$ is the index set of the buses equipped with PVs. $p_i^{PV}$, $q_i^{PV}$, and $s_i$ are the active power, reactive power, and the capacity of PV at bus $i$, respectively. In this paper, each PV inverter is oversized with $s_i = 1.2\, p_{i,\max}, \forall i \in V^{PV}$. The maximum PV reactive power is $q_{i,\max}^{PV} = \sqrt{s_i^2 - (p_i^{PV})^2}$. Note that the objective of the OPF problem is equivalent to minimize the overall power loss.

Eq.5 may be infeasible due to the large penetration of PVs. In this case, slack variables can be added on the voltage constraint.

## A.4 Droop Control

The droop control, as recommended by IEEE Std-1547™-2018 [7], follows the control strategy $q_i^{PV} = f(v_i)$ where $q_i^{PV}$ and $v_i$ are the PV reactive power and the voltage measurement of a PV bus $i$. $f(\cdot)$ is piecewise linear as shown in Figure 2. In detail, $v_{\text{ref}}$ represents the voltage set point (e.g. 1.0 p.u.). $v_{\text{a}}$ and $v_{\text{d}}$ represent the saturation regions limited by the PV inverter capacity and the current PV active power. There also exists a dead-band between $v_{\text{b}}$ and $v_{\text{c}}$ that does not require any control. For the voltage lower than $v_{\text{b}}$, the inverter provides reactive power proportional to the voltage deviation against $v_{\text{ref}}$. If the voltage is higher than $v_{\text{c}}$, the inverter absorbs reactive power until convergence achieves. The droop control only requires the local voltage measurements that is simple and efficient to implement. However, it cannot directly minimise the power losses nor respond to fast voltage changes. For simplicity, we set $v_{\text{b}} = v_{\text{c}} = v_{\text{ref}}$ in this work.

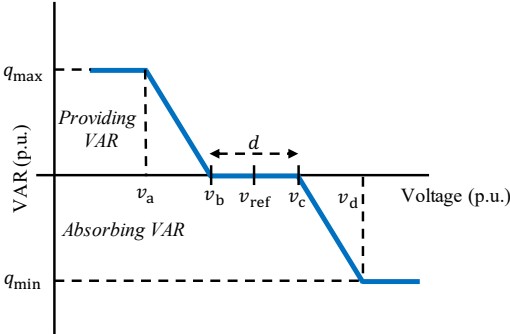

Figure 2: The illustration of the droop control law.

## B  COMA with Continuous Actions

COMA [10] is an MARL algorithm with credit assignments over Q-value functions via the mechanism of counterfactual regret, however, it can only serve for the discrete action space. In this paper, to enable COMA eligible for the continuous action space, we conduct some tiny adjustments on the construction of Q-value for each agent. The original version of calculating each agent's Q-value assignment w.r.t. the discrete actions is shown as follows:

$$Q_i(\mathbf{s}, \mathbf{a}) = Q(\mathbf{s}, \mathbf{a}) - \sum_{a_i' \in \mathcal{A}_i} \pi_i(a_i' | \tau_i) Q(\mathbf{s}, \mathbf{a}_{-i}, a_i'), \tag{6}$$

where $\tau_i$ is a history of agent $i$; $\mathbf{a}_{-i} = \times_{j \neq i} a_j$. To fit the continuous actions, we simply change Eq.6 to the form such that

$$Q_i(\mathbf{s}, \mathbf{a}) = Q(\mathbf{s}, \mathbf{a}) - \int_{a_i' \in \mathcal{A}_i} Q(\mathbf{s}, \mathbf{a}_{-i}, a_i') \, d\pi_i(a_i' | \tau_i), \tag{7}$$

where $\pi_i(a_i' | \tau_i)$ is a Gaussian distribution over $a_i'$. In practice, $\int_{a_i' \in \mathcal{A}_i} Q(\mathbf{s}, \mathbf{a}_{-i}, a_i') \, d\pi_i(a_i' | \tau_i)$ is approximated via Monte Carlo sampling, so it can be written as follows:

$$Q_i(\mathbf{s}, \mathbf{a}) = Q(\mathbf{s}, \mathbf{a}) - \frac{1}{M} \sum_{k=1}^{M} Q(\mathbf{s}, \mathbf{a}_{-i}, (a_i')_k), \quad (a_i')_k \sim \pi_i(a_i' | \tau_i). \tag{8}$$

## C  Experimental Settings

The source code of experimentation will be released after acceptance of paper for the easy reproductions and further studies.

## C.1 Algorithm Settings and Training Details

Since IDDPG and MADDPG do not possess any extra hyperparameters, we only introduce the hyperparameters of COMA, MATD3, SQDDPG, IPPO, and MAPPO that we used in experiments.

**Common Settings.** All algorithms are trained with online learning (i.e., for the on-policy algorithms like IPPO and MAPPO the behaviour policies are updated once at the end of each episode; for the on-policy algorithm like COMA the behaviour policies/values are updated every 60 time steps; and for the off-policy algorithms like SQDDPG, IDDPG and MADDPG the behaviour policies/values are updated every 60 time steps, where all data used for training are collected online) and the target policy/critic networks are updated every interval that is twice as the update interval of behaviour policy/critic introduced above. Taking the lessons from [11, 12], the algorithms except for MAPPO and IPPO update critic networks with 10 epochs while update policy networks with 1 epoch. All algorithms are trained with the normalised reward and the action bound enforcement trick [13] that works better than the hard clipping in our experiments. The target update learning rate is set to $0.1$. The gradient is clipped with L1 norm and the clip bound is set to $1$. The batch size of training data is set to 32 and the replay buffer size for off-policy algorithms is set to $5,000$. Agent ID is concatenated with the observation and the layer normalisation [14] is applied to the first layer after the observation input. The parameters are shared among agents in this experiment. As for the policy network, RNN with GRUs [15] is applied as a filter to solve the partial observation problems. The critic network is constructed with pure MLPs. The general settings of the policy and critic networks are shown in Table 1. During training, a fixed standard deviation as $1.0$ is applied to conduct the exploration. For the policy loss with entropy, the entropy penalty is set to 1e-3. The parameter initialisation is implemented by sampling from normal distribution with $\mathcal{N}(0, 0.1)$. RMSProp [16] is used as the optimizer, with the learning rate of 1e-4 for updating both policies and critics.

**COMA.** The sample size M of COMA for continuous actions proposed in this paper is set to 10 in experiments.

**MATD3.** The clip boundary c for clipping the exploration noise is set to 1 in experiments.

**SQDDPG.** The sample size M of SQDDPG is set to 10 in experiments.

**IPPO and MAPPO.** We apply generalised advantage estimation (GAE) [17] to evaluate the return with $\lambda = 0.95$. The value loss coefficient is set to 2. The $\epsilon$ for clipping the objective function is set to 0.4. We also normalise the advantages during training to reduce variance. 10 epochs of training are conducted for each time of update.

All hyperparameters reported above are tuned by the grid search and the best ones are selected as the final choices.

Table 1: The general specifications for policy and value networks.

| NETWORK | STRUCTURE |
|---|---|
| POLICY | LINEAR(STATE_DIM, 64) $\to$ LAYERNORM() $\to$ RELU() $\to$ GRU(64, 64) $\to$ LINEAR(64, ACTION_DIM) |
| CRITIC | LINEAR(INPUT_DIM, 64) $\to$ LAYERNORM() $\to$ RELU() $\to$ LINEAR(STATE_DIM, 64) $\to$ RELU() $\to$ LINEAR(64, OUTPUT_DIM) |

In addition to the training details introduced in the main part of paper, we expose more in this section. At the initial state we randomly sample reactive power of generators so that the experiments are more realistic, i.e. to test whether the agents can solve any emergent situations. The training time varies from 2.5 to 4 hrs, dependent on the selection of scenarios and algorithms.

## C.2 Process of Simulations

We show the flow chart in Figure 3 to illustrate the process of the execution of the simulation for distributed active voltage control on power distribution networks. At the beginning of each episode, a series of consecutive PV and load profiles for 480 time steps (i.e. 1 day) is in buffer. For each time step, the relevant PV and load profile are extracted, combined with the voltage status computed by Pandapower [18] (i.e., the power flow is calculated) to form the next state. Additionally, the reward is also calculated according to the results computed by Pandapower. Before fed to agents, the received state will be split to a batch of observations as per the region where each agent is located. Each

agent only receives a local observation and the global reward, then it makes next decision. The above procedure is repeated until the end of an episode.

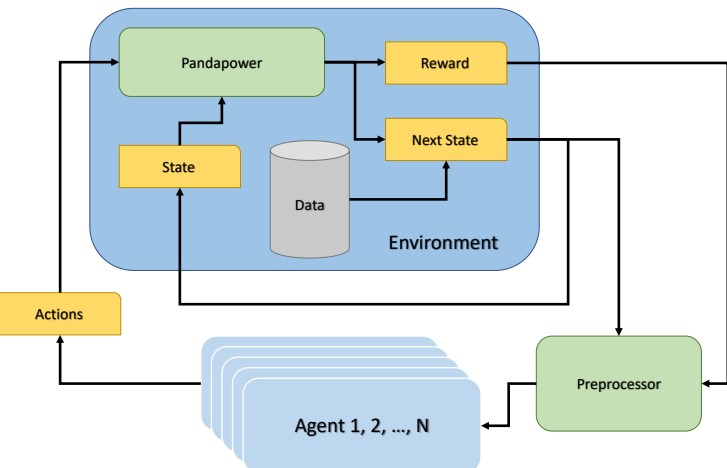

Figure 3: The flow chart of the implementation of environment for distributed active voltage control on power distribution networks.

### C.3 Voltage Barrier Functions

In this paper, we compare 3 different voltage barrier functions applied in this work. The L1-shape can be written as follows:

$$l_v(v_k) = |v_k - v_{\text{ref}}|, \quad \forall k \in V. \tag{9}$$

The L2-shape can be written as follows:

$$l_v(v_k) = (v_k - v_{\text{ref}})^2, \quad \forall k \in V. \tag{10}$$

The Bowl-shape can be written as follows:

$$l_v(v_k) = \begin{cases} a \cdot |v_k - v_{\text{ref}}| - b & \text{If } |v_k - v_{\text{ref}}| > 0.05, \\ -c \cdot \mathcal{N}(v_k \mid v_{\text{ref}}, 0.1) + d & \text{Otherwise,} \end{cases} \tag{11}$$

where $a, b, c, d$ are 4 hyperparameters to adjust the shape and smoothness of function that are set to $2, 0.095, 0.01, 0.04$ respectively in this work; $\mathcal{N}(v_k \mid v_{\text{ref}}, 0.1)$ is a density function for the normal distribution with the mean as $v_{\text{ref}}$ and the standard deviation as 0.1. In addition to the significance of satisfying the objective of active voltage control, this construction can also be interpreted as a sort of statistical implication. $v_k$ is assumed to follow the Laplace distribution outside the safety range while it is assumed to follow the normal distribution inside the safety range. Thereafter, the active voltage control problem can be transformed to the maximum likelihood estimation (MLE) over a mixture distribution of voltage with a constraint on reactive power generations.

### C.4 Reward for the Safety of Power Networks

Although the action range has been restricted to avoid the violence of the loading capacity of power networks, in experiments there still exists possibilities that this accident could happen. To resolve this problem, if the violence of the loading capacity appears, the system would backtrack to the last state as well as terminate the simulation and give a penalty of $-200$ as the reward instead of the one shown in the main part of paper.

## D Environmental Settings

We present 3 MV/LV distribution network models, each of which is composed of distinct topology and parameters, a load profile (including both active and reactive powers) describing different user

behaviours, and a PV profile describing the active power generation from PVs. Although it is possible to partition the control regions by the voltage sensitivity of each bus [19], they are commonly determined by different distribution network owners in practice. Consequently, the control regions in this paper are partitioned by the shortest path between the coupling bus and the terminal bus. Besides, each region consists of 1-4 PVs depending on the zonal sizes.

## D.1 Network Topology

A summary of the 3 networks is recorded in Table 2 and the specific topologies are demonstrated in Figure 4.

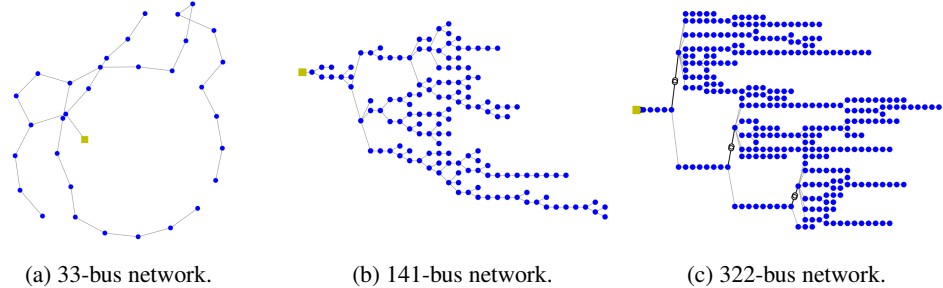

(a) 33-bus network.    (b) 141-bus network.    (c) 322-bus network.

Figure 4: Topologies of power networks. The yellow square is the reference bus (a.k.a. the slack bus) and each blue circle is a non-reference bus. Transformers are highlighted as double-circles.

Table 2: Network specifications

|        | Rated Voltage  | No. Loads | No. Regions | No. PVs | $p_{\max}^{L}$ | $p_{\max}^{PV}$ |
|--------|----------------|-----------|-------------|---------|----------------|-----------------|
| 33-bus | 12.66 kV       | 32        | 4           | 6       | 3.5 MW         | 8.75 MW         |
| 141-bus| 12.5 kV        | 84        | 9           | 22      | 20 MW          | 80 MW           |
| 322-bus| 110-20-0.4 kV  | 337       | 22          | 38      | 1.5 MW         | 3.75 MW         |

**33-bus.** The 33-bus network is modified from case33bw in MATPOWER [20] and PandaPower [18]. To guarantee the tree structure, similar to [21], we drop lines 33-37 to avoid any loops. 6 PVs are added unevenly on bus 13 and 18 (zone 1), bus 22 (zone 2), bus 25 (zone 3), bus 29 and 33 (zone 4). The PV-load ratio is $PR = 2.5$.

**141-bus.** The 141-bus network is modified from case141 in MATPOWER [20] as well. A similar procedure is followed as 33-bus network.

**322-bus.** The proposed 322-bus network consists of an external 110-kV bus, a long medium-voltage (20 kV) line (25 buses in total) and 3 LV feeders (0.4 kV) representing rural (128 buses), semi-urban (110 buses), and urban (58 buses) areas defined by SimBench [22]. Areas with different voltage levels are connected though standard transformers defined in PandaPower [18]. The rural area has the lowest power consumption level and some buses are with no loads, while more than one load are allowed to locate on a bus in the urban area, so the total number of loads is higher than the number of buses in the 322-bus network. The users can also generate their own synthetic networks by following out procedure. To simplify the settings, we aggregate the multiply loads at each bus into one.

## D.2 Data Descriptions.

**Load Profiles.** The load profile of each network is modified based on the real-time Portuguese electricity consumption accounting for 232 consumers of 3 years[1]. The original dataset contains 370 residential and industrial clients electricity usage from 2011 to 2014 with 15-min resolution. As some of the data does not start at the beginning, we collect the data from 2012-01-01 00:15:00 and

---

[1] https://archive.ics.uci.edu/ml/datasets/ElectricityLoadDiagrams20112014.

delete the locations that contain more than 20 missing data. The remaining missing data (mostly because of the winter time to daylight saving time switch) is interpolated linearly. The load data is then interpolated with 3-min resolution which is consistent with the real-time control period in the grid. The final data size is $526080 \times 232$ accounting for load profiles for 232 consumers of 1096 days (three years). We then remove the outliers that are outside $7\sigma$ against the mean value. For 33-bus and 141-bus networks, the 232 load profiles are randomly assigned to each bus. For 322-bus network, repeated load profiles are allowed. In practice, Gaussian noises are added to load active and reactive powers.

**PV Profiles.** Ten cities/regions/provinces in Belgium, Netherlands, and Luxembourg are considered to represent the distinct zonal solar radiation levels, including Antwerp, Brussels, Flemish-Brabant (a province of Belgium), Hainaut (a province of Belgium), Liege, Limburg (a province of Netherland), Luxembourg, Namur, Walloon-Brabant (a province of Belgium), and West-Flanders (a province of Belgium). The PV data is collected from Elia group[2], a Belgiums power network operator. The PV data is also interpolated with 3-min resolution resulting in $526080 \times 10$ data in total. For 33-bus (with 4 regions) and 141-bus (with 9 regions) networks, PV profiles are randomly assigned to each region. For 322-bus (with 22 regions) system, different regions can have the same PV profiles. Note that the PVs in the same control region share the same PV profiles as they are geometrically contiguous. In real-time, we also add Gaussian noise to the PV active power.

We summarise the load and PV profiles of different scales in Figure 5-11 below.

Figure 5 illustrates the total PV active power generation and active load consumption in 33-bus network. Figure 6 illustrates four distinct PV buses in 33-bus network of January and July. Note that bus-13 and bus-18 are in the same region, so they have the same PV profiles.

Figure 7 illustrates the total PV active power generation and active load consumption in 141-bus network. Figure 8 illustrates four distinct PV buses in 141-bus network of January and July. Note that bus-36 and bus-111 are in the same region, so they have the same PV profiles.

Figure 9 illustrates the total PV active power generation and active load consumption in 322-bus network. Figure 10 illustrates four distinct PV buses in 322-bus network of January and July.

Figure 11 illustrates the power factors (PFs) of the three systems under test. Higher power factors ($> 0.9$) usually represents the residential consumers while low power factors ($< 0.5$) can represent the industrial consumers.

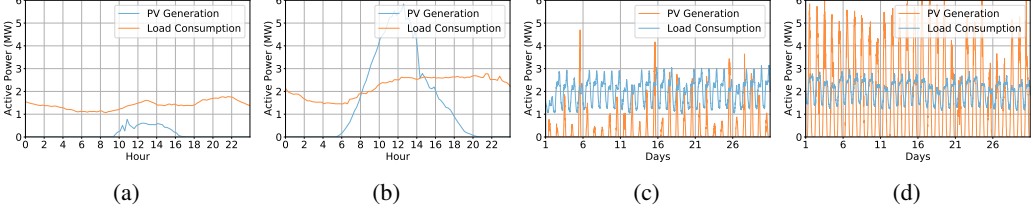

(a)  (b)  (c)  (d)

Figure 5: Total power of 33-bus network: (a): a winter day, (b): a summer day, (c): a winter month (January), (d): a summer month (July)

# E   Extra Experimental Results

## E.1   Extra Results during Training

In addition to the control rate (CR) and power loss (PL) introduced in the main part of paper, we also introduce 2 extra metrics to evaluate the performances of algorithms during training.

- *Voltage out of control ratio (VR)*: It calculates the average of the ratio of voltage outside the safety range (i.e. 0.95-1.05 $p.u.$) per time step during an episode.
- *Q loss (QL)*: It calculates the average of the mean reactive power generations by agents per time step during an episode.

---

[2]https://www.elia.be/en/grid-data/power-generation/solar-pv-power-generation-data.

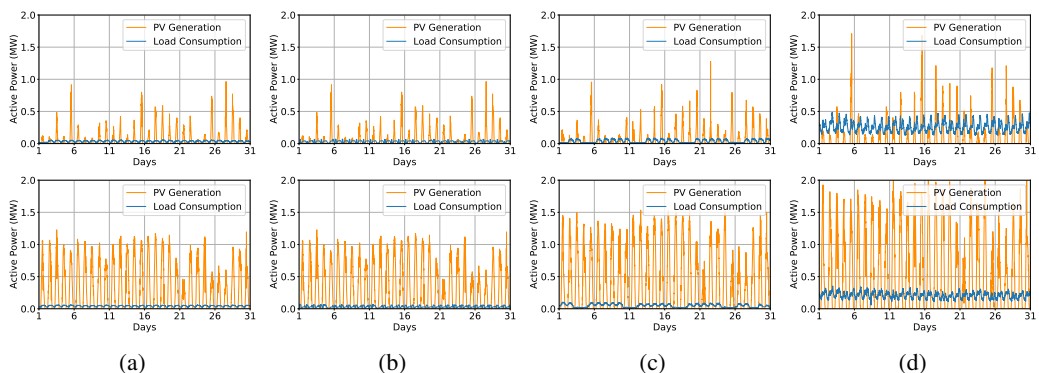

(a)        (b)        (c)        (d)

Figure 6: Daily power of 33-bus network: active PV power generation and active load consumption for different buses in 33-bus network. (a): bus-13, (b): bus-18, (c): bus-22 (d): bus-25. The first row: power in a winter month (January), second row: power in a summer month (July).

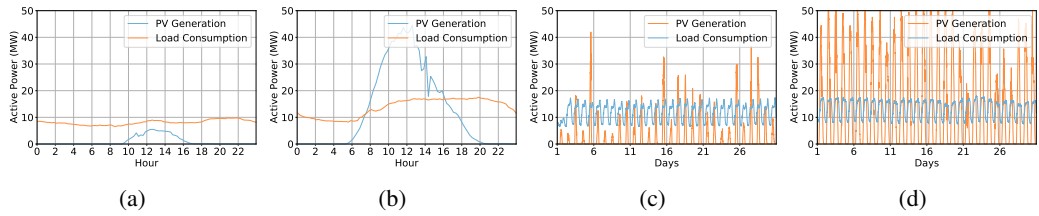

(a)        (b)        (c)        (d)

Figure 7: Total power of 141-bus network: (a): a winter day, (b): a summer day, (c): a winter month (January), (d): a summer month (July)

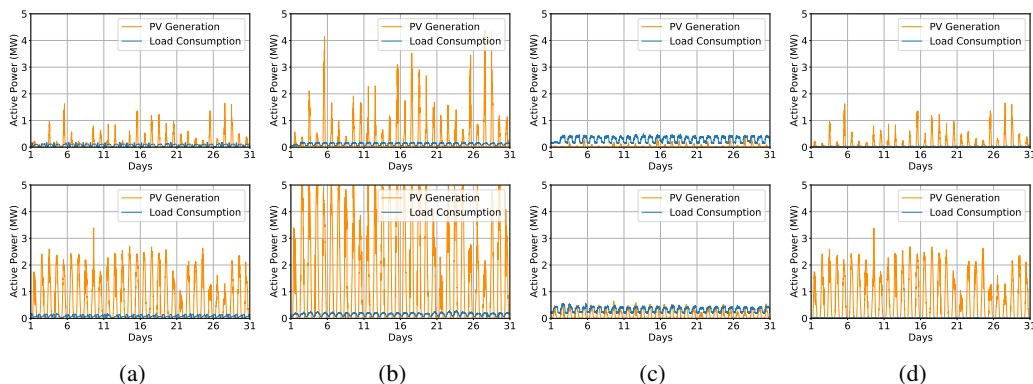

(a)        (b)        (c)        (d)

Figure 8: Daily power of 141-bus network: active PV power generation and active load consumption for different buses in 141-bus network. (a): bus-36, (b): bus-77, (c): bus-100 (d): bus-111. The first row: power in winter month (January), second row: power in summer month (July).

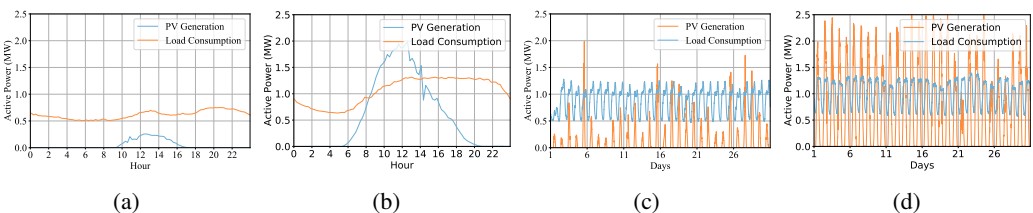

(a)        (b)        (c)        (d)

Figure 9: Total power of 322-bus network: (a): a winter day, (b): a summer day, (c): a winter month (January), (d): a summer month (July)

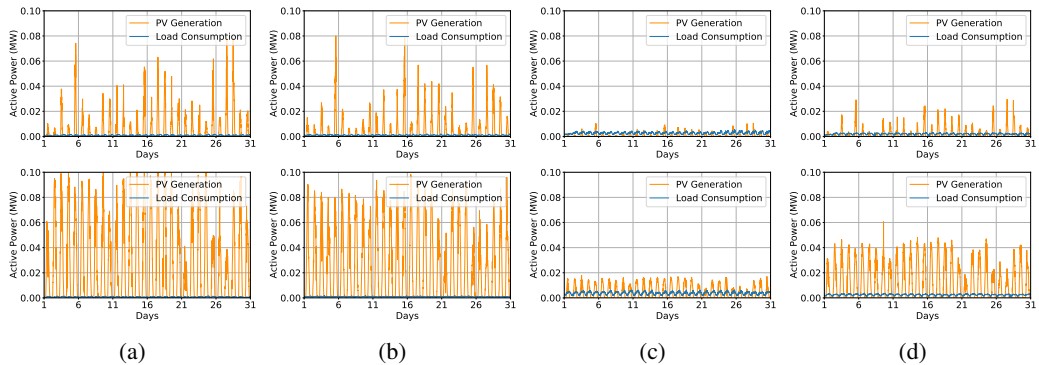

(a)      (b)      (c)      (d)

Figure 10: Daily power of 322-bus network: active PV power generation and active load consumption for different buses in 322-bus network. (a): bus-54, (b): bus-147, (c): bus-297 (d): bus-322. The first row: power in winter month (January), second row: power in summer month (July).

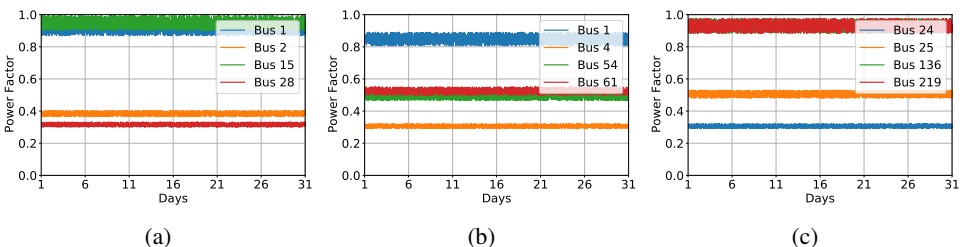

(a)          (b)          (c)

Figure 11: Power factors of four buses in (a): 33-bus, (b): 141-bus, and (c): 322-bus networks

Similar to the results before, all performances are measured by the median metrics of 5 random seeds and each test is conducted by 10 randomly selected episodes.

## E.2 Extra Results for Case Studies

In this section, we show more results on the case studies for the comparison between MARL and the traditional control methods (i.e. OPF and droop control). For 141-bus network, MATD3 trained by Bowl-shape voltage barrier function acts as the candidate for MARL. For 322-bus network, MADDPG trained by L2-shape voltage barrier function acts as the candidate for MARL.

**One Bus in 141-Bus Network.** Figure 14 shows the results for a typical bus in the 141-bus network. In summer, all methods can control the voltage within the safety range in most of time, except that MARL fails to control the voltage from 20:00 to 22:00. Nonetheless, the power loss of MARL is lower than the droop control. In winter, all methods can control the voltage within the safety range, however, MARL behaves exclusively on generating the reactive power, i.e. generating more reactive power, while the power loss of MARL is still lower than that of the droop control.

**One Bus in 322-Bus Network.** Figure 15 shows the results for a typical bus in the 322-bus network. In summer, only droop control can control the voltage within the safety range. MARL and OPF cannot control the voltage within the safety range from 10:00 to 14:00 when the PV active power is extremely high. The bad performance of OPF is possibly due to the reason that the 322-bus network is so large and complicated that it may suffer the computational catastrophe w.r.t. the inverse of the topological matrix. In winter, all methods can control the voltage within the safety range, though the voltage at this time is originally within the safety range without no control. It is so strange that MARL decrease the voltage so that it is near the lower bound of the safety range. Apparently, the strategy of MARL for this case is suboptimal. The additional penetration of reactive power by MARL induces the excessive power loss. The intrinsic reason of this phenomenon deserves to be investigated in the future work.

**Analysis for All Buses.** To give the whole picture of active voltage control for the days we selected for demonstrations above, we show the status of all buses in Figure 16 for the 33-bus network, Figure

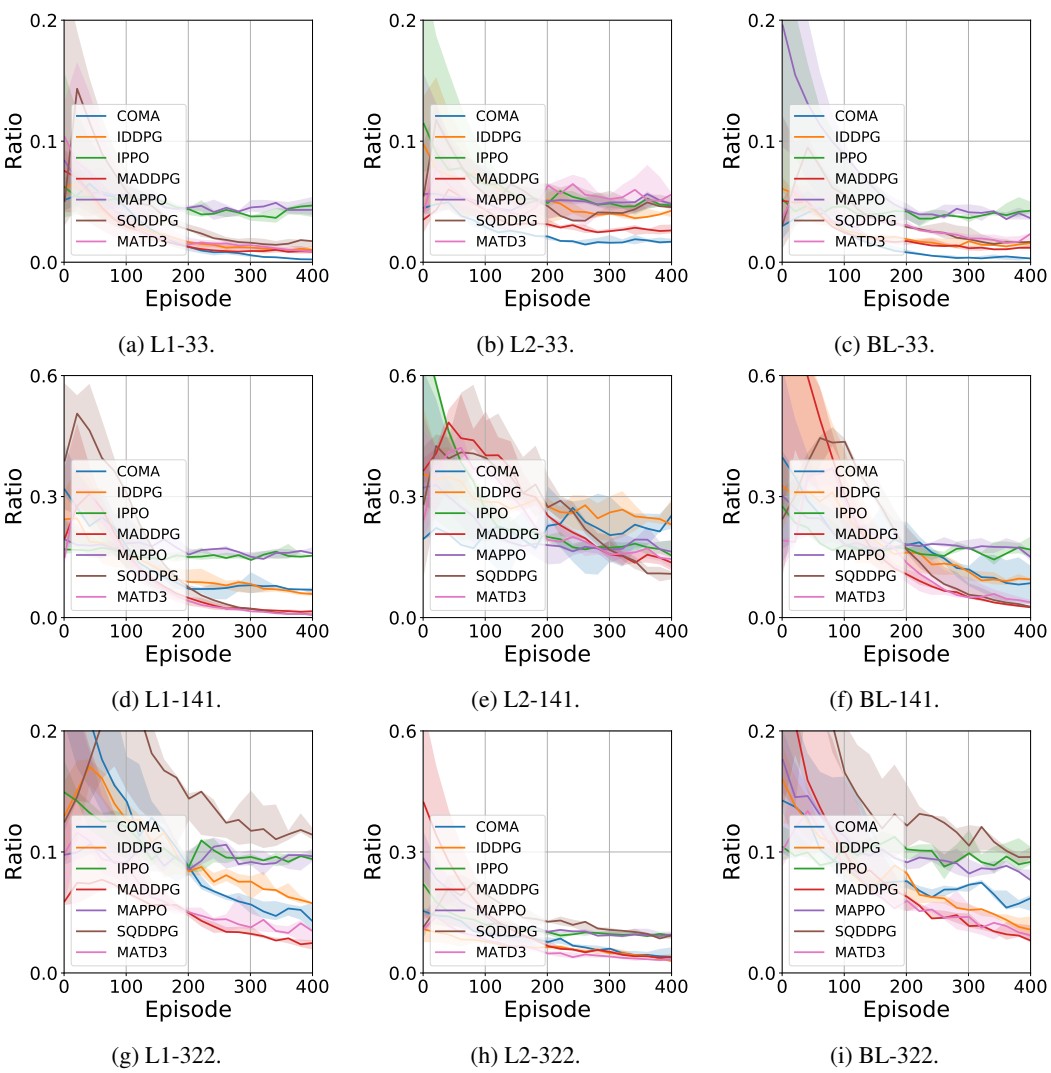

Figure 12: Voltage out of control ratio of algorithms with different reward functions consisting of distinct voltage barrier functions. The sub-caption indicates barrier-scenario and BL is the contraction of Bowl.

17 for the 141-bus network and Figure 18 for the 322-bus network. In winter, all methods can control the voltages of all buses within the safety range whatever the scenario is. We just focus on the results for summer in the following discussion. For the 33-bus network and the 141-bus network, it is obvious that the traditional control methods can control the voltage within the safety range, while MARL loses control on some buses from 18:00 to 24:00. This implies that MARL may tend to learn solving the the situations of high PV penetrations. The possible reason could be that the situations of high penetrations appear more frequently, which leads to a known problem existing in MARL called relative overgeneralisation [23]. For the 322-bus network, the performance of droop control is far better than OPF and MARL. The reason for the failure of OPF is the computational burden as we stated before. It is worth noting that droop control relies on a high-bandwidth inner loop in inverter controller so the effective control rate is much higher than the sample rate [24].

## E.3 Extra Results during Testing

To show the results more convincingly, we also report the mean test results on the final models of MARL after training. The tests on each algorithm are repeated with 10 randomly selected initial states. Noticeably, we report both mean and standard deviation to exhibit the randomness of 10 tests

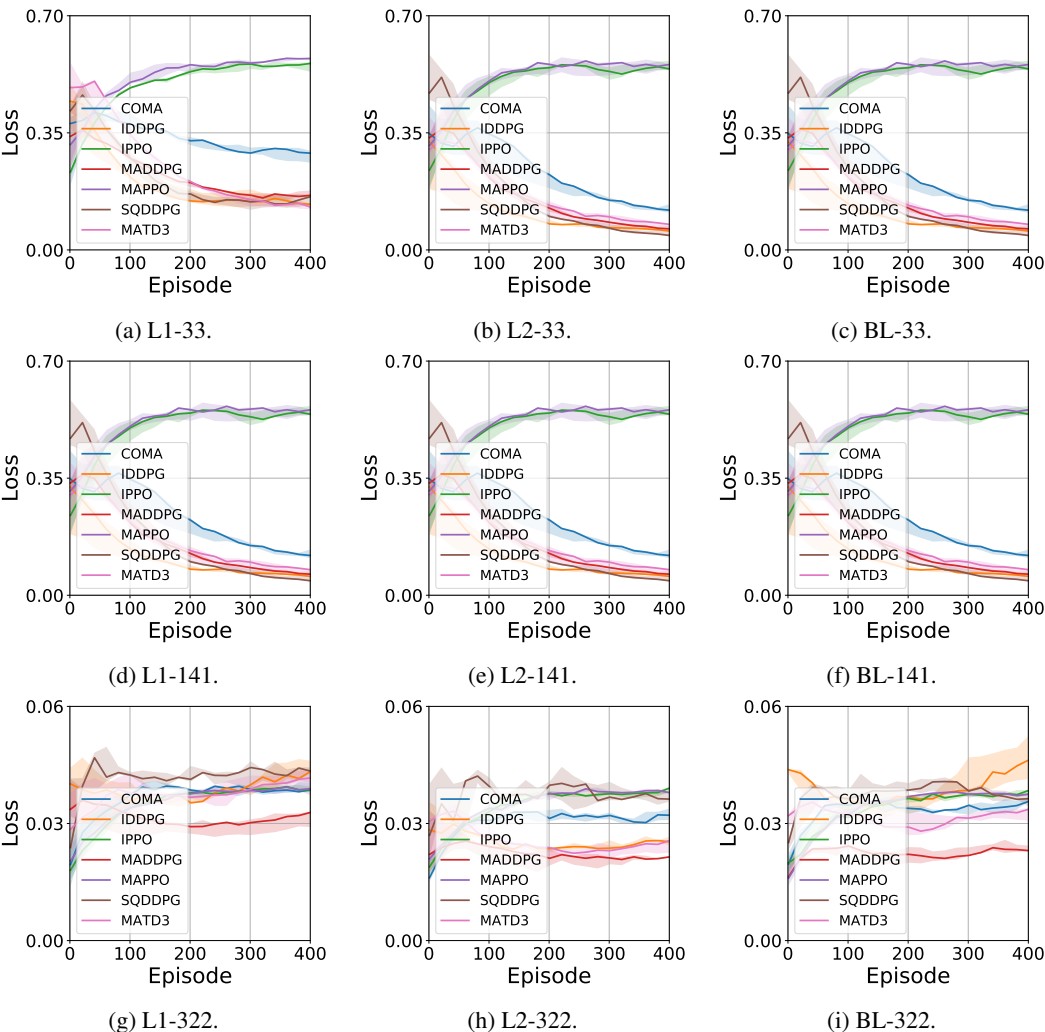

Figure 13: Q (reactive power) losses of algorithms with different reward functions consisting of distinct voltage barrier functions. The sub-caption indicates barrier-scenario and BL is the contraction of Bowl.

for the metrics of continuous values. Since the metrics of ratio does not satisfy the hypothesis of normality, we just report the the mean of 10 tests for appropriateness. Additionally, we also report the results of the traditional control methods with 100 randomly selected episodes.

We now introduce the metrics used in the Table 3-5.

- *% V. Out of Control*: The average of the ratio of the voltages out of control per time step during each episode.

- *% V. below* $0.95v_{ref}$: The average ratio of the voltages below $0.95v_{ref}$ per time step during each episode.

- *% V. above* $1.05v_{ref}$: The average ratio of the voltages above $1.05v_{ref}$ per time step during each episode.

- *% CR*: The ratio of time steps where all buses' voltages being under control during each episode.

- *V. Dev.*: The average voltage deviations (away from the $v_{ref}$) during each episode.

- *Max V. Drop Dev.*: The average of the maximum deviation of voltage drop (i.e. below $0.95v_{ref}$) per time step during each episode.

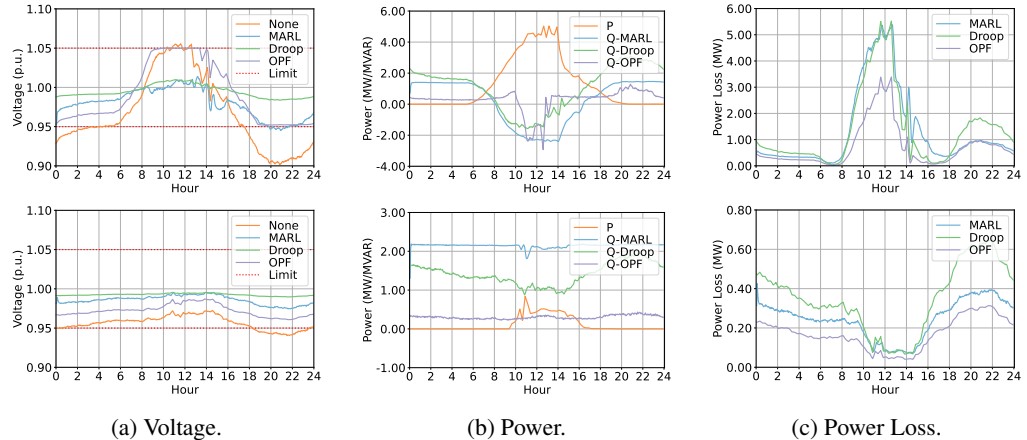

(a) Voltage.      (b) Power.      (c) Power Loss.

Figure 14: Compare MARL with traditional control methods on a typical bus during a day for the 141-bus network. 1st row: results for summer. 2nd row: results for winter. None and limit in (a) represent the voltage with no control and the safety range respectively. P and Q in (b) indicate the PV active power and the reactive power by various methods.

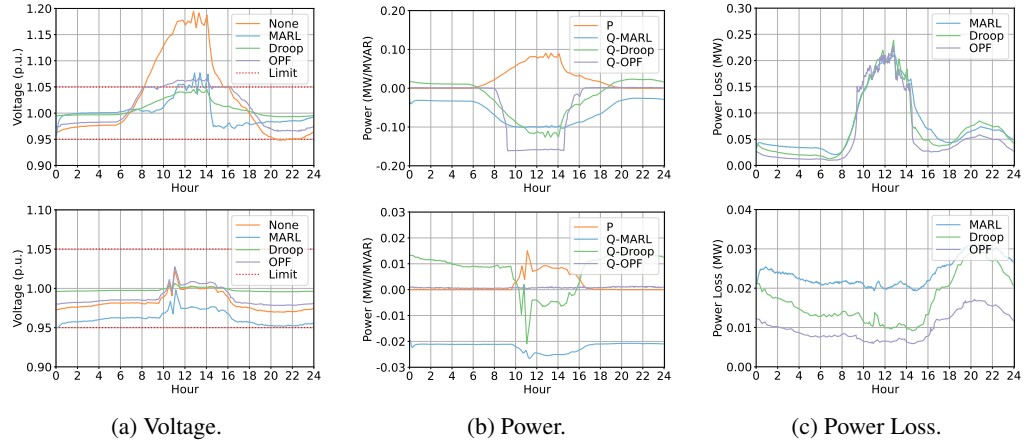

(a) Voltage.      (b) Power.      (c) Power Loss.

Figure 15: Compare MARL with traditional control methods on a typical bus during a day for the 322-bus network. 1st row: results for summer. 2nd row: results for winter. None and limit in (a) represent the voltage with no control and the safety range respectively. P and Q in (b) indicate the PV active power and the reactive power by various methods.

- *Max V. Rise Dev.*: The average of the maximum deviation of voltage rise (i.e. above $1.05v_{\text{ref}}$) per time step during each episode.
- *PL*: The average of the total power loss over all buses per time step dyuring each episode.

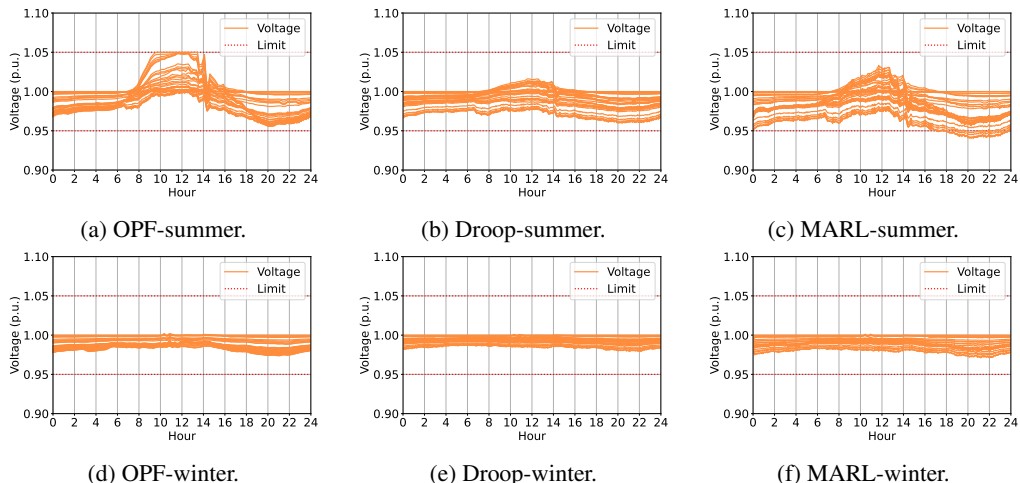

Figure 16: The status of all buses for a day on 33-bus network. The green lines are the variation of the voltage of buses and red dashed line is the safety boundary. Each caption above indicates method-season.

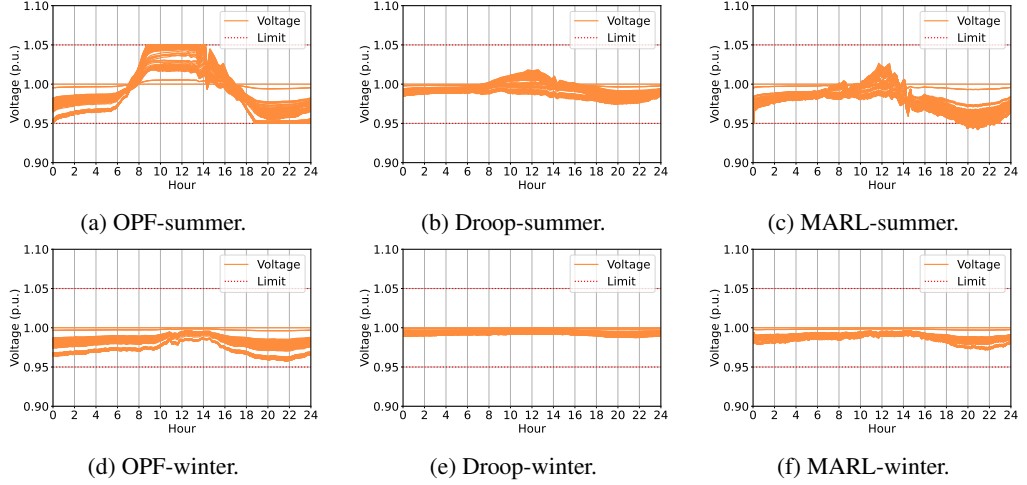

Figure 17: The status of all buses for a day on 141-bus network. The green lines are the variation of the voltage of buses and red dashed line is the safety boundary. Each caption above indicates method-season.

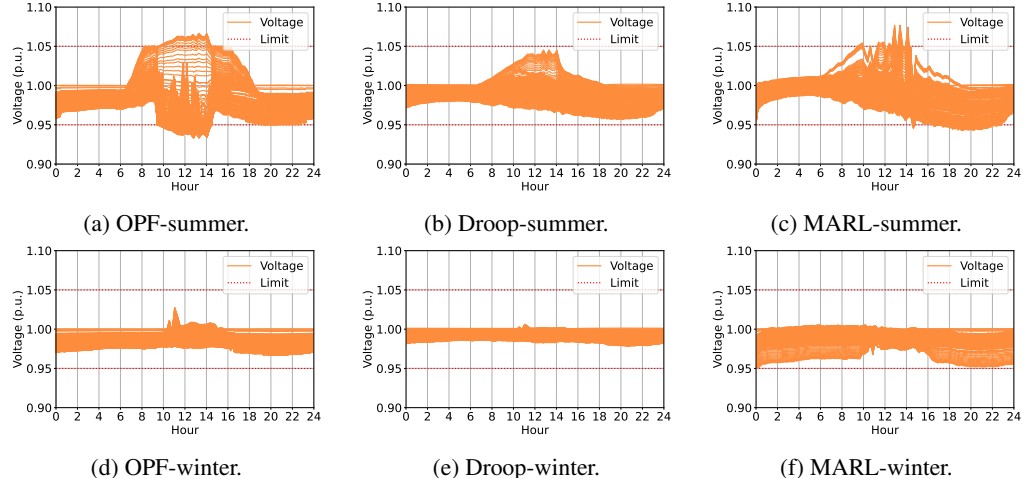

Figure 18: The status of all buses for a day on 322-bus network. The green lines are the variation of the voltage of buses and red dashed line is the safety boundary. Each caption above indicates method-season.

Table 3: The mean test results on the 33-bus network with 10 randomly selected episodes for MARL and 100 random selected episodes for the traditional control methods. The results are recorded with mean (±std.).

| METHOD | % V. OUT OF CONTROL | % V. BELOW | % V. ABOVE | % CR | V. DEV. | MAX V. DROP DEV. | MAX V. RISE DEV. | PL |
|---|---|---|---|---|---|---|---|---|
| NO CONTROL | 6.3 | 5.1 | 1.2 | 70.6 | 0.021 ± 0.005 | 0.036 ± 0.011 | 0.010 ± 0.015 | 0.069 ± 0.036 |
| DROOP CONTROL | 0.0 | 0.0 | 0.0 | 100.0 | 0.011 ± 0.002 | 0.025 ± 0.006 | 0.003 ± 0.004 | 0.082 ± 0.064 |
| OPF | 0.0 | 0.0 | 0.0 | 100.0 | 0.014 ± 0.003 | 0.020 ± 0.009 | 0.011 ± 0.013 | 0.056 ± 0.046 |
| IDDPG-L1 | 1.1 | 0.7 | 0.4 | 90.3 | 0.014 ± 0.002 | 0.000 ± 0.000 | 0.001 ± 0.000 | 0.066 ± 0.007 |
| MADDPG-L1 | 0.9 | 0.4 | 0.5 | 92.0 | 0.013 ± 0.000 | 0.000 ± 0.000 | 0.001 ± 0.000 | 0.071 ± 0.004 |
| COMA-L1 | 0.2 | 0.0 | 0.2 | 97.0 | 0.011 ± 0.000 | 0.000 ± 0.000 | 0.000 ± 0.000 | 0.105 ± 0.002 |
| IPPO-L1 | 4.5 | 0.0 | 4.5 | 68.1 | 0.015 ± 0.001 | 0.000 ± 0.000 | 0.008 ± 0.002 | 0.148 ± 0.010 |
| MAPPO-L1 | 4.5 | 0.0 | 4.5 | 68.2 | 0.015 ± 0.001 | 0.008 ± 0.002 | 0.008 ± 0.002 | 0.154 ± 0.005 |
| MATD3-L1 | 1.0 | 0.8 | 0.2 | 91.6 | 0.015 ± 0.001 | 0.000 ± 0.000 | 0.000 ± 0.000 | 0.064 ± 0.004 |
| SQDDPG-L1 | 1.5 | 0.8 | 0.3 | 87.2 | 0.015 ± 0.001 | 0.000 ± 0.000 | 0.001 ± 0.000 | 0.068 ± 0.005 |
| IDDPG-L2 | 4.4 | 3.5 | 0.9 | 75.4 | 0.020 ± 0.001 | 0.001 ± 0.000 | 0.001 ± 0.001 | 0.067 ± 0.005 |
| MADDPG-L2 | 2.8 | 2.0 | 0.8 | 79.7 | 0.018 ± 0.000 | 0.001 ± 0.000 | 0.001 ± 0.000 | 0.063 ± 0.004 |
| COMA-L2 | 1.7 | 0.9 | 0.8 | 85.8 | 0.015 ± 0.001 | 0.000 ± 0.000 | 0.001 ± 0.000 | 0.068 ± 0.004 |
| IPPO-L2 | 4.5 | 0.1 | 4.5 | 72.0 | 0.015 ± 0.001 | 0.000 ± 0.000 | 0.009 ± 0.001 | 0.142 ± 0.008 |
| MAPPO-L2 | 4.7 | 0.0 | 4.7 | 70.9 | 0.015 ± 0.001 | 0.000 ± 0.000 | 0.010 ± 0.001 | 0.139 ± 0.005 |
| MATD3-L2 | 5.4 | 4.9 | 0.5 | 75.0 | 0.021 ± 0.001 | 0.002 ± 0.001 | 0.001 ± 0.000 | 0.074 ± 0.004 |
| SQDDPG-L2 | 4.9 | 3.9 | 1.1 | 74.0 | 0.021 ± 0.001 | 0.001 ± 0.000 | 0.002 ± 0.001 | 0.071 ± 0.003 |
| IDDPG-BL | 1.5 | 1.2 | 0.3 | 87.0 | 0.015 ± 0.000 | 0.012 ± 0.003 | 0.003 ± 0.003 | 0.069 ± 0.007 |
| MADDPG-BL | 1.2 | 1.0 | 0.2 | 89.0 | 0.015 ± 0.001 | 0.010 ± 0.003 | 0.002 ± 0.001 | 0.073 ± 0.004 |
| COMA-BL | 0.3 | 0.2 | 0.2 | 96.5 | 0.011 ± 0.000 | 0.002 ± 0.001 | 0.002 ± 0.001 | 0.106 ± 0.005 |
| IPPO-BL | 4.6 | 0.1 | 4.5 | 65.4 | 0.015 ± 0.001 | 0.001 ± 0.001 | 0.045 ± 0.008 | 0.154 ± 0.013 |
| MAPPO-BL | 3.8 | 0.1 | 3.7 | 70.6 | 0.014 ± 0.001 | 0.001 ± 0.001 | 0.037 ± 0.006 | 0.150 ± 0.008 |
| MATD3-BL | 2.2 | 2.0 | 0.2 | 84.0 | 0.019 ± 0.001 | 0.020 ± 0.002 | 0.002 ± 0.001 | 0.077 ± 0.009 |
| SQDDPG-BL | 1.8 | 1.1 | 0.7 | 85.4 | 0.016 ± 0.001 | 0.011 ± 0.005 | 0.007 ± 0.002 | 0.072 ± 0.007 |

Table 4: The mean test results on the 141-bus network with 10 randomly selected episodes for MARL and 100 random selected episodes for the traditional control methods. The results are recorded with mean (±std.).

| METHOD | % V. OUT OF CONTROL | % V. BELOW | % V. ABOVE | % CR | V. DEV. | MAX V. DROP DEV. | MAX V. RISE DEV. | PL |
|---|---|---|---|---|---|---|---|---|
| NO CONTROL | 32.3 | 23.8 | 8.6 | 37.9 | $0.042 \pm 0.008$ | $0.046 \pm 0.021$ | $0.016 \pm 0.022$ | $0.956 \pm 0.610$ |
| DROOP CONTROL | 0.0 | 0.0 | 0.0 | 100.0 | $0.009 \pm 0.002$ | $0.014 \pm 0.003$ | $0.003 \pm 0.004$ | $1.519 \pm 1.335$ |
| OPF | 0.0 | 0.0 | 0.0 | 100.0 | $0.021 \pm 0.006$ | $0.020 \pm 0.008$ | $0.011 \pm 0.012$ | $0.819 \pm 0.922$ |
| IDDPG-L1 | 9.7 | 7.7 | 2.0 | 71.9 | $0.026 \pm 0.002$ | $0.003 \pm 0.002$ | $0.001 \pm 0.000$ | $1.167 \pm 0.137$ |
| MADDPG-L1 | 2.4 | 1.1 | 1.3 | 92.3 | $0.016 \pm 0.002$ | $0.000 \pm 0.000$ | $0.001 \pm 0.000$ | $1.525 \pm 0.137$ |
| COMA-L1 | 8.9 | 7.2 | 1.6 | 73.9 | $0.023 \pm 0.007$ | $0.004 \pm 0.004$ | $0.001 \pm 0.001$ | $1.639 \pm 0.216$ |
| IPPO-L1 | 13.8 | 0.0 | 13.8 | 77.3 | $0.026 \pm 0.002$ | $0.000 \pm 0.000$ | $0.011 \pm 0.002$ | $1.380 \pm 0.061$ |
| MAPPO-L1 | 16.0 | 0.1 | 15.9 | 74.6 | $0.028 \pm 0.002$ | $0.000 \pm 0.000$ | $0.013 \pm 0.002$ | $1.465 \pm 0.069$ |
| MATD3-L1 | 2.3 | 0.4 | 1.9 | 94.1 | $0.015 \pm 0.001$ | $0.000 \pm 0.000$ | $0.001 \pm 0.001$ | $1.608 \pm 0.107$ |
| SQDDPG-L1 | 1.8 | 0.7 | 1.0 | 96.0 | $0.015 \pm 0.001$ | $0.001 \pm 0.000$ | $0.001 \pm 0.000$ | $1.757 \pm 0.064$ |
| IDDPG-L2 | 26.0 | 20.1 | 5.9 | 49.3 | $0.038 \pm 0.003$ | $0.008 \pm 0.002$ | $0.004 \pm 0.001$ | $0.966 \pm 0.085$ |
| MADDPG-L2 | 12.6 | 9.3 | 3.3 | 68.9 | $0.028 \pm 0.002$ | $0.003 \pm 0.001$ | $0.002 \pm 0.000$ | $1.007 \pm 0.098$ |
| COMA-L2 | 26.6 | 16.1 | 10.5 | 41.4 | $0.038 \pm 0.005$ | $0.016 \pm 0.011$ | $0.012 \pm 0.005$ | $1.989 \pm 0.369$ |
| IPPO-L2 | 16.7 | 0.2 | 16.5 | 72.9 | $0.028 \pm 0.003$ | $0.000 \pm 0.000$ | $0.013 \pm 0.003$ | $1.418 \pm 0.129$ |
| MAPPO-L2 | 17.1 | 0.1 | 17.0 | 72.6 | $0.029 \pm 0.002$ | $0.000 \pm 0.000$ | $0.014 \pm 0.002$ | $1.472 \pm 0.043$ |
| MATD3-L2 | 14.6 | 11.3 | 3.3 | 63.9 | $0.030 \pm 0.003$ | $0.004 \pm 0.001$ | $0.002 \pm 0.001$ | $0.954 \pm 0.063$ |
| SQDDPG-L2 | 14.3 | 10.2 | 4.2 | 67.1 | $0.029 \pm 0.007$ | $0.006 \pm 0.006$ | $0.003 \pm 0.002$ | $1.350 \pm 0.257$ |
| IDDPG-BL | 8.8 | 7.2 | 1.6 | 74.3 | $0.025 \pm 0.003$ | $0.003 \pm 0.001$ | $0.001 \pm 0.000$ | $1.136 \pm 0.110$ |
| MADDPG-BL | 2.5 | 1.6 | 0.9 | 91.6 | $0.020 \pm 0.001$ | $0.001 \pm 0.000$ | $0.001 \pm 0.000$ | $1.350 \pm 0.088$ |
| COMA-BL | 9.3 | 6.8 | 2.5 | 72.0 | $0.024 \pm 0.005$ | $0.004 \pm 0.004$ | $0.002 \pm 0.002$ | $1.954 \pm 0.413$ |
| IPPO-BL | 17.5 | 0.1 | 17.4 | 72.2 | $0.029 \pm 0.002$ | $0.000 \pm 0.000$ | $0.014 \pm 0.003$ | $1.450 \pm 0.052$ |
| MAPPO-BL | 15.8 | 0.1 | 15.7 | 75.0 | $0.028 \pm 0.003$ | $0.000 \pm 0.000$ | $0.013 \pm 0.003$ | $1.500 \pm 0.133$ |
| MATD3-BL | 5.8 | 4.3 | 1.4 | 81.3 | $0.021 \pm 0.005$ | $0.002 \pm 0.003$ | $0.001 \pm 0.000$ | $1.313 \pm 0.086$ |
| SQDDPG-BL | 2.9 | 1.5 | 1.5 | 92.1 | $0.016 \pm 0.001$ | $0.001 \pm 0.000$ | $0.001 \pm 0.000$ | $1.720 \pm 0.235$ |

Table 5: The mean test results on the 322-bus network with 10 randomly selected episodes for MARL and 100 random selected episodes for the traditional control methods. The results are recorded with mean (±std.).

| METHOD | % V. OUT OF CONTROL | % V. BELOW | % V. ABOVE | % CR | V. DEV. | MAX V. DROP DEV. | MAX V. RISE DEV. | PL |
|---|---|---|---|---|---|---|---|---|
| NO CONTROL | 18.2 | 15.6 | 2.5 | 32.1 | $0.032 \pm 0.006$ | $0.052 \pm 0.013$ | $0.028 \pm 0.035$ | $0.038 \pm 0.017$ |
| DROOP CONTROL | 0.0 | 0.0 | 0.0 | 99.4 | $0.011 \pm 0.002$ | $0.027 \pm 0.005$ | $0.008 \pm 0.008$ | $0.061 \pm 0.043$ |
| OPF | 5.0 | 4.6 | 0.3 | 86.8 | $0.015 \pm 0.008$ | $0.026 \pm 0.010$ | $0.018 \pm 0.014$ | $0.057 \pm 0.036$ |
| IDDPG-L1 | 6.1 | 1.5 | 4.6 | 36.7 | $0.018 \pm 0.002$ | $0.007 \pm 0.004$ | $0.015 \pm 0.006$ | $0.109 \pm 0.013$ |
| MADDPG-L1 | 2.3 | 0.1 | 2.2 | 77.7 | $0.013 \pm 0.001$ | $0.000 \pm 0.000$ | $0.009 \pm 0.002$ | $0.070 \pm 0.007$ |
| COMA-L1 | 4.7 | 0.2 | 4.5 | 56.6 | $0.017 \pm 0.002$ | $0.000 \pm 0.000$ | $0.016 \pm 0.004$ | $0.091 \pm 0.008$ |
| IPPO-L1 | 9.3 | 0.1 | 9.2 | 39.1 | $0.024 \pm 0.001$ | $0.000 \pm 0.000$ | $0.037 \pm 0.005$ | $0.103 \pm 0.003$ |
| MAPPO-L1 | 9.8 | 0.1 | 9.7 | 40.0 | $0.024 \pm 0.001$ | $0.000 \pm 0.000$ | $0.039 \pm 0.004$ | $0.102 \pm 0.004$ |
| MATD3-L1 | 3.2 | 1.3 | 1.9 | 64.8 | $0.015 \pm 0.001$ | $0.004 \pm 0.003$ | $0.008 \pm 0.002$ | $0.078 \pm 0.007$ |
| SQDDPG-L1 | 11.7 | 0.3 | 11.4 | 29.6 | $0.026 \pm 0.001$ | $0.001 \pm 0.001$ | $0.042 \pm 0.006$ | $0.117 \pm 0.013$ |
| IDDPG-L2 | 3.6 | 0.9 | 2.7 | 65.9 | $0.016 \pm 0.001$ | $0.001 \pm 0.001$ | $0.011 \pm 0.004$ | $0.070 \pm 0.010$ |
| MADDPG-L2 | 3.8 | 0.6 | 3.2 | 67.3 | $0.016 \pm 0.001$ | $0.001 \pm 0.000$ | $0.015 \pm 0.003$ | $0.053 \pm 0.004$ |
| COMA-L2 | 4.9 | 0.2 | 4.7 | 51.1 | $0.017 \pm 0.002$ | $0.000 \pm 0.000$ | $0.016 \pm 0.006$ | $0.081 \pm 0.007$ |
| IPPO-L2 | 9.1 | 0.1 | 9.0 | 39.7 | $0.024 \pm 0.001$ | $0.000 \pm 0.000$ | $0.036 \pm 0.004$ | $0.103 \pm 0.004$ |
| MAPPO-L2 | 10.0 | 0.1 | 9.9 | 40.5 | $0.024 \pm 0.002$ | $0.000 \pm 0.000$ | $0.040 \pm 0.006$ | $0.100 \pm 0.007$ |
| MATD3-L2 | 3.1 | 0.5 | 2.6 | 71.1 | $0.015 \pm 0.001$ | $0.001 \pm 0.000$ | $0.012 \pm 0.003$ | $0.058 \pm 0.007$ |
| SQDDPG-L2 | 9.1 | 0.3 | 8.8 | 44.0 | $0.023 \pm 0.001$ | $0.000 \pm 0.000$ | $0.036 \pm 0.004$ | $0.097 \pm 0.007$ |
| IDDPG-BL | 4.7 | 1.1 | 3.6 | 40.9 | $0.017 \pm 0.003$ | $0.005 \pm 0.005$ | $0.012 \pm 0.004$ | $0.128 \pm 0.024$ |
| MADDPG-BL | 3.2 | 0.9 | 2.3 | 67.7 | $0.016 \pm 0.001$ | $0.001 \pm 0.001$ | $0.011 \pm 0.005$ | $0.065 \pm 0.008$ |
| COMA-BL | 5.7 | 0.2 | 5.5 | 43.2 | $0.017 \pm 0.001$ | $0.001 \pm 0.001$ | $0.018 \pm 0.004$ | $0.098 \pm 0.010$ |
| IPPO-BL | 9.3 | 0.1 | 9.2 | 41.0 | $0.024 \pm 0.001$ | $0.000 \pm 0.000$ | $0.037 \pm 0.005$ | $0.101 \pm 0.003$ |
| MAPPO-BL | 8.2 | 0.1 | 8.1 | 44.6 | $0.022 \pm 0.001$ | $0.000 \pm 0.000$ | $0.032 \pm 0.004$ | $0.097 \pm 0.003$ |
| MATD3-BL | 2.8 | 0.8 | 2.0 | 68.1 | $0.016 \pm 0.001$ | $0.001 \pm 0.001$ | $0.010 \pm 0.003$ | $0.074 \pm 0.006$ |
| SQDDPG-BL | 9.7 | 0.2 | 9.5 | 40.6 | $0.024 \pm 0.001$ | $0.000 \pm 0.000$ | $0.039 \pm 0.003$ | $0.100 \pm 0.012$ |