# OpenReview forum: "Multi-Agent Reinforcement Learning for Active Voltage Control on Power Distribution Networks"
_NeurIPS.cc/2021/Conference — NeurIPS 2021 Poster_

### Official Review · Reviewer_6BFo · 2021-06-28

**Rating:** 5
**Confidence:** 4

**Summary:**

The main contribution of the article is mention in the abstract "... establishes an open-source environment. It aims to bridge the gap between the power community and the MARL community". and In section 1: "There is no commonly accepted benchmark to provide the basis for fair comparison of different solutions. To facilitate further research on this topic, and to bridge the gap between the power community and MARL community", The authors seem to forget about L2RPN.

**Limitations And Societal Impact:**

They need to be more elaborate about the limitations of the proposed technique/framework, secondly they should clearly mention the limitations of their solution within the proposed framework.


**Main Review:**

- The authors say in section 1 "we present a comprehensive test-bench and open-source environment for MARL based active voltage control". The effort for standardization is not good enough because they do not seem to know what measures should be taken to this. If there has to be a standard framework for RL based learning of smart grids then it must include the following specifications
    - Definition of actions and their domains
    - Definition of states and effect of grid-users on the states
    - Definition of environment and its standard response to actions
    - Define standards for randomness and noise in states, observations, actions, and their effects.
- The algorithms ànd networks need to be fully customizable.


- Formulation is sketchy. The actions are only dependent on the reactive power, while other researchers have used many different types of actions. Restricting the algorithm to limited actions will definitely affect the results, but the authors do not seem to realize that. In section 5.3 they describe the difference of results between their algorithms and traditional solutions, but they seem to forget that they are not using the complete range of actions here.

- The authors seem to forget that the network layers matter a lot with respect to the algorithm, see (https://openreview.net/forum?id=LmUJqB1Cz8)

- The authors also seem to have forgotten the effect of replay selection on the results. I conclude this by their choice reward function which they name as bowl function, which could also be a parabolic function too. Probably this is why they have doubts about their statements e.g. line 302-303, and section 5.3.

- The implementation details mentioned in the appendix do not add much to the article.



**Time Spent Reviewing:**

3

---

> ### Author Response · Authors · 2021-08-10
> **Response to reviewer 6BFo**
>
> ### We thank the reviewer for the thoughtful comments
>
> 1. > The main contribution of the article is mention in the abstract "... establishes an open-source environment. It aims to bridge the gap between the power community and the MARL community" and in section 1: "There is no commonly accepted benchmark to provide the basis for fair comparison of different solutions. To facilitate further research on this topic, and to bridge the gap between the power community and MARL community". The authors seem to forget about L2RPN.
>
>    ***Response:*** The authors do notice Grid2Op L2RPN and its success in running the L2PRN challenge. However, the core of Grid2Op is not compatible with widely used RL and MARL environments, such as openAI gym and Whiteson's MARL framework. Although it can be interfaced to other environments, the extra interfacing layer may introduce addtional computations in trainning and add barriers in tuning. That is, the gap still exists. This paper refomulates the environment in a way that is completely compatibable with Whiteson's MARL framework and therefore reduces the barriers for MARL researchers to get into the area. Compared with Grid2Op, our framework also features more transparent data structures to facilitate more efficient training and tunning.
>
>    Moreover, this paper addresses a different problem compared to L2PRN. L2PRN addresses the power flow control problem in tranmission networks, and this paper addresses the active voltage control problems in distribution networks. The active voltage control problem is related to a vast number of distributed generators which is decentralised in nature, in contrast to the centralised policy in L2PRN. This further justified why L2PRN is not adequate a new framework is needed.
>
> 2. > The authors say in section 1 "we present a comprehensive test-bench and open-source environment for MARL based active voltage control". The effort for standardization is not good enough because they do not seem to know what measures should be taken to this. If there has to be a standard framework for RL based learning of smart grids then it must include the following specifications
>    > * Definition of actions and their domains
>    > * Definition of states and effect of grid-users on the states
>    > * Definition of environment and its standard response to actions
>    > * Define standards for randomness and noise in states, observations, actions, and their effects.
>
>    ***Response:*** Standardisation is not the target of this paper although it is our long-term ambition. The motivation of this paper is to formulate the voltage control problem and create an environment in a way that can be easily understood and used by MARL researchers.
>
> 3. > The algorithms and networks need to be fully customizable.
>
>    ***Response:*** The authors agree with the reviewer's point and our enviroment is indeed customisable in both algorithms and networks.
>
> 4. > Formulation is sketchy. The actions are only dependent on the reactive power, while other researchers have used many different types of actions. Restricting the algorithm to limited actions will definitely affect the results, but the authors do not seem to realize that. In section 5.3 they describe the difference of results between their algorithms and traditional solutions, but they seem to forget that they are not using the complete range of actions here.
>
>     ***Response:***  The authors agree that there are multiple choices of actions for power system control, but for the active voltage control problem the most reasonable choice of actions is the reactive power of distributed generators. It is also posible to create an inner loop (such as droop control) so that an agent controls voltage by seting the reference voltage instead of controlling reactive power directly. However, this is not allowed by current grid codes of most countries and is not inline with the end-to-end philosophy of machine learning. Most literature on active voltage control chooses reactive power as the actions. See references below.
>
> 5. > The authors seem to forget that the network layers matter a lot with respect to the algorithm, see (https://openreview.net/forum?id=LmUJqB1Cz8)
>
>    ***Response:*** It is not clear what is meant by "network layer", but the choice of using reactive power for voltage control has been justified in the response to review point 4.
>
> 6. > The authors also seem to have forgotten the effect of replay selection on the results. I conclude this by their choice reward function which they name as bowl function, which could also be a parabolic function too. Probably this is why they have doubts about their statements e.g. line 302-303, and section 5.3.
>
>    ***Response:*** Replay selection can be considered. However, we cannot see the relationship between replay and our proposed bowl function. The introduction of bowl function and other barrier functions is to address the disappearance of gradient when the power flow is beyond the physical limit.
>
>
> 7. > The implementation details mentioned in the appendix do not add much to the article
>
>    ***Response:*** This is due to the limitation of the pages. For this reason, the authors only give a general logic and main experimental results, which is the common style for ML papers.
>
> ***Reference***
>
> [1] Wang, Qi, et al. "Two-stage voltage control strategy for PV plants based on variable droop control." International Journal of Electronics 107.2 (2020): 250-271.
>
> [2] Yang, Qiuling, et al. "Two-timescale voltage control in distribution grids using deep reinforcement learning." IEEE Transactions on Smart Grid 11.3 (2019): 2313-2323.
>
> [3] Cao, D., Zhao, J., Hu, W., Ding, F., Huang, Q., Chen, Z., & Blaabjerg, F. (2021). Data-Driven Multi-agent Deep Reinforcement Learning for Distribution System Decentralized Voltage Control with High Penetration of PVs. IEEE Transactions on Smart Grid.
>
> [4] Liu, H., & Wu, W. (2021). Online multi-agent reinforcement learning for decentralized inverter-based volt-var control. IEEE Transactions on Smart Grid.
>
> [5] Agalgaonkar, Y. P., Pal, B. C., & Jabr, R. A. (2013). Distribution voltage control considering the impact of PV generation on tap changers and autonomous regulators. IEEE Transactions on Power Systems, 29(1), 182-192.
>
> [6] Singhal, A., Ajjarapu, V., Fuller, J., & Hansen, J. (2018). Real-time local volt/var control under external disturbances with high PV penetration. IEEE Transactions on Smart Grid, 10(4), 3849-3859.
>
> [7] "IEEE Standard for Interconnection and Interoperability of Distributed Energy Resources with Associated Electric Power Systems Interfaces," in IEEE Std 1547-2018 (Revision of IEEE Std 1547-2003) , vol., no., pp.1-138, 6 April 2018, doi: 10.1109/IEEESTD.2018.8332112.
>
> [8] Yoon, D., Hong, S., Lee, B. J., & Kim, K. E. (2020, September). Winning the L2RPN Challenge: Power Grid Management via Semi-Markov Afterstate Actor-Critic. In International Conference on Learning Representations.
>
> [9] Masters, C. L. (2002). Voltage rise: the big issue when connecting embedded generation to long 11 kV overhead lines. Power engineering journal, 16(1), 5-12.
>
> [10] Zhang, B., Lam, A. Y., Domínguez-García, A. D., & Tse, D. (2014). An optimal and distributed method for voltage regulation in power distribution systems. IEEE Transactions on Power Systems, 30(4), 1714-1726.

---

> > ### Comment · Reviewer_6BFo · 2021-08-18
> > **Further explanation of point 5 and 6**
> >
> > Point 5: I want to point out that the type of Neural network you use in the RL setting can change the results to a great deal. The learnability of the whole system also depends upon the type of Neural network you are using
> >
> > Pint 6: It is hard to explain in few lines, but the problem of vanishing gradient and exploding gradient are both tied to the replay selection. Yes, you can control them using reward function, but a better replay selection can also solve the problem. I am sorry I am forgetting the reference at this time.

---

> > > ### Author Response · Authors · 2021-08-18
> > > **Further Reply to point 5 and 6**
> > >
> > > 1. > I want to point out that the type of Neural network you use in the RL setting can change the results to a great deal. The learnability of the whole system also depends upon the type of Neural network you are using
> > >
> > > ***Response:*** Yes, we agree with this point. However, in this paper we just provide an initial results for evaluating the environment to give some insights and hints for the future work by MARL community. The more attempts on neural network structures can be conducted in the future work.
> > >
> > > 2. > Pint 6: It is hard to explain in few lines, but the problem of vanishing gradient and exploding gradient are both tied to the replay selection. Yes, you can control them using reward function, but a better replay selection can also solve the problem. I am sorry I am forgetting the reference at this time.
> > >
> > > ***Response:*** Yes, replay buffer may affect the results. Actually, there are some other factors that may also affect the problem of vanishing gradient and exploding gradient, e.g., activation function, layer normalisation, etc.. Nevertheless, in this paper we focus on the reward function **that is the fundamental element correlated with the general RL itself** (though we agree that replay buffer is also a key component, but it only affects off-policy algorithms). The rest of factors deserve to be studied in the future.

---

> > > ### Comment · Reviewer_S8R6 · 2021-08-24
> > > **Clarification of setting**
> > >
> > > I'd like to emphasize the authors' point that the multi-agent distribution system voltage control setting (what they address) is very different from the single-agent transmission system topology switching + dispatch setting (what L2RPN addresses).
> > >
> > > Further, the points on NN, replay, etc. that Reviewer 6BFo makes are not incorrect, but are in my opinion irrelevant to the evaluation of the submission. The value of this submission is in providing an environment, and the MARL models they use to evaluate this environment are just an initial demonstration.

---

### Official Review · Reviewer_DvqK · 2021-07-16

**Rating:** 6
**Confidence:** 3

**Summary:**

This paper introduces a new open-source environment for training RL agents on controlling the voltage of a power distribution network. The problem is formulated as a multi-agent Dec-POMDP. Agents control PV inverters using only local measurements coming from sensors situated in the region they operate. Their task is to maintain the voltage within safety range. Several MARL algorithms are evaluated in three different power networks. The paper also investigates different reward functions.

**Main Review:**

The main contribution of this work is the introduction of a new benchmark domain for testing MARL algorithms. This consists of a simulator of a power network. The environment is built using existent network topologies and real data. I definitely agree with the authors' view on the importance of having realistic scenarios where RL methods can be tested appropriately before using them in real-world applications. It is very unfortunate that the code is not provided in the supplementary material. It is hard to assess how useful this work will be for the community without being able to test the environment.

I miss some intuitive explanation of the dynamics of the problem. Some technical details are given in the Appendix but these are hard to follow for someone lacking in knowledge of the field. In particular, I am looking for details like, how much past information about the network state is useful for making decisions. How do the actions taken in the past affect the network in the long run. I.e. what's the horizon of the problem? All this could be illustrated using some sample trajectories coming from the standard operation of the network. To what extent do the changes in the policies of the agents in one region affect other regions? I.e how strongly coupled are the different regions in the network? This information could shed some light on why some MARL algorithms seem to perform better than others.

It is not clear whether an agent controls one or multiple PV inverters. Why are there multiple agents for each region? As far as I can see, all agents in a region receive the same information. If that’s the case, wouldn’t it be more effective to have all PV inverters within a region be controlled by a single policy?

The environment is partially observable, yet according to Section 5: MARL Algorithm Settings (line 283-287) the policies are modeled with MLPs receiving only the last two observations as input. Using only the last two observations doesn’t seem sufficient to satisfy the Markov assumption, especially considering the intrinsic non-stationarity of the problem (power generation and consumption fluctuate during the day).

Minor details:

h is first defined as a function that maps nodes to the corresponding subset of measures for that node (line 195) and then also used as the history length (line 228).



**Time Spent Reviewing:**

5

---

> ### Author Response · Authors · 2021-08-10
> **Response to reviewer DvqK**
>
> ### We thank the reviewer for insightful comments
>
> 1. > It is very unfortunate that the code is not provided in the supplementary material. It is hard to assess how useful this work will be for the community without being able to test the environment.
>
>    ***Response:*** The authors promises that the source code will be released if the paper is accepted. The authors will also write a specific document to illustrate the usage of the environment.
>
> 2. > I miss some intuitive explanation of the dynamics of the problem. Some technical details are given in the Appendix but these are hard to follow for someone lacking in knowledge of the field. In particular, I am looking for details like, how much past information about the network state is useful for making decisions. How do the actions taken in the past affect the network in the long run. I.e. what's the horizon of the problem? All this could be illustrated using some sample trajectories coming from the standard operation of the network. To what extent do the changes in the policies of the agents in one region affect other regions? I.e how strongly coupled are the different regions in the network? This information could shed some light on why some MARL algorithms seem to perform better than others.
>
>    ***Response:*** (1) The intuition of the dynamics can be explained as the change of voltage, power and loads must satisfy a power balance. Under this power balance, the problem aims to take the control of reactive power under evolved loads to enable the voltage of all buses to be within a safety range. (2) In experiments, the authors currently feed the agent two-step past observation, but did several trials. The authors conducted experiments and found that two-step past observation is more efficient than one-step. If the agent is fed with the history observations with more steps, the effect does not improve. Since the interval between two time steps is 15 mins and the horizon considered is an interval with consecutive 24 hours that is equal to 96 steps but with a random initial time. To ease life, the horizon with 100 steps is used during training. (3) Yes, the policies of the agents in one region could affect the other regions. The strength depends on the connectivity and topology of the network as well as the reaactive power the agent generates. Another reason for MARL performing better is the coordination between regions as the reviewer thought of. However, in real world how to award the operators (agents) with those excessive reactive power and how to punish the operators that generate less reactive power to support the region where they belong to is an open question. For this reason, the problem considered in this paper is difficult and needs much further research on it.
>
> 3. > It is not clear whether an agent controls one or multiple PV inverters. Why are there multiple agents for each region? As far as I can see, all agents in a region receive the same information. If that’s the case, wouldn’t it be more effective to have all PV inverters within a region be controlled by a single policy?
>
>    ***Response:*** First, most of the existing works consider the decentralised scenario, i.e., all PV inverters controlled by a single policy. If each agent is an operator, then it means that each region is monopolized by only an operator. Although it seems effective as the reviewer said, the rights of customers to choose the operator are taken. On the other hand, the monopolization is not healthy for the electricity market, e.g., the quality of service could become worse. To address all of these limitations, in this paper we consider the distributed scenario, where each agent (i.e. an operator) controls a PV inverter. As a result, in a region the users can select favourite operators. As for the share of information in a region, it can be seen as a market rule over operators within the region to keep the safety of the operations. If each agent only observe the local information on its bus (node), it is usually difficult to control the voltage within the safety range and guarantee the service quality. The current academics for studying the smart grid of renewable energies are all based on the assumptions, since these research outcomes are for the future deployment.
>
> 4. > The environment is partially observable, yet according to Section 5: MARL Algorithm Settings (line 283-287) the policies are modeled with MLPs receiving only the last two observations as input. Using only the last two observations doesn’t seem sufficient to satisfy the Markov assumption, especially considering the intrinsic non-stationarity of the problem (power generation and consumption fluctuate during the day).
>
>    ***Response:*** The authors think this question is highly correlated with the effectiveness of past information in the previous questions and agree with the reviewer's point. The authors also did the experiments with RNN after the submission and will add it to the final manuscript if accepted. Interestingly, the authors find that the results of RNN-based policies are more stable but with no much improvement on the final performance. The intrinsic reason deserves to be studied in the future work. Thanks the reviewer for the catch.

---

> > ### Comment · Reviewer_DvqK · 2021-08-30
> > **Response to Authors**
> >
> > Thanks for your clarifications.
> >
> > Unfortunately, I cannot increase my score based only on the authors' promise that they will include all the necessary information in the final version. I believe the concerns I raised in my previous message to be very important and thus, the new version will need to be properly reviewed before it can be published.

---

> > > ### Author Response · Authors · 2021-08-30
> > > **Response to Reviewer**
> > >
> > > Thanks for your comments. We understand the concerns from you, however, we promise the detailed information of usage for the environment will be given.

---

### Official Review · Reviewer_S8R6 · 2021-07-22

**Rating:** 6
**Confidence:** 4

**Summary:**

This paper proposes a new environment for multi-agent reinforcement learning that addresses the problem of active voltage control on electric power distribution networks. The authors formulate the voltage control problem as a Dec-POMDP, and implement a number of different MARL baselines to assess their performance on the environment. They also assess the effect of different voltage loss functions on this environment.

**Limitations And Societal Impact:**

Yes, the authors have adequately addressed these points.

**Main Review:**

I am very much on the borderline regarding this paper. On the one hand, I think it addresses a very important problem, and that it would be extremely beneficial to both the ML/MARL and power systems communities to have this kind of distribution system control environment readily available. On the other hand, I think one of the key assumptions underpinning the current framework needs to be better justified, that there should be more details about the actual usability of the environment, and that some of the technical notations and explanations need to be cleaned up.

Key assumption: One of my main concerns is regarding the assumption made in the paper that multiple inverters are controlled by one company that has full information within its operational region. I do not believe this model is common in present distribution networks -- that is, where smart inverter control exists, it is generally the case that each inverter operates separately based on local information. The authors should provide much more context around this assumption (e.g., how does this grid operation mode relate to existing grid operation modes?), both to avoid inadvertently misleading the ML community about the state of current grid operations, and to better justify the technical assumptions on which the rest of the work is based.

Usage instructions: Since the proposed contribution is an environment with baselines, I was surprised to see no details relating to the practical usability of the environment. For instance, would this environment be integrated within existing MARL frameworks, and/or does it have a user interface that is similar? Is the plan to provide a quickstart guide to help users easily get started? These kinds of details are important to understanding whether this kind of environment would actually be able to be widely adopted, or whether ML researchers/practitioners would find it difficult to use in practice.

Terminology and explanations: As one of the goals of the paper is to make the voltage control setting accessible to ML practitioners, the authors should be careful to go through and make sure all power systems terminology and assumptions are properly defined, e.g.,:
* Line 92: Define "power curtailment"
* Line 127--129: It would be important to clarify why and when the distribution network can be modeled as a tree graph (especially since the 33-bus system evaluated is actually not a tree, though the authors mention in the appendix that they modify it to remove loops).
* Line 140: Define "per unit"

Other questions and comments:
* Lines 41--42: Voltage and power are indeed related, through the power flow equations, which means that the constraint and control action are indeed linked. Would it be possible to clarify what is meant by this statement?
* Line 140: "When the load is heavy during the nighttime" -- Why is load heavier at night, especially during seasons with no HVAC? Or is it just meant that *net* load is heavy due to low availability of PV power at night?
* Line 184--186: The current notation for $\mathcal{R}$ implies that it is a set of sets, rather than a union of sets.
* Line 191: What is meant by "at the last step"?
* Line 205, "State Transition Probability Function": The notation here should be cleaned up, as $\delta$ outputs a state but $Pr(s_{t+1}|s_{t})$ outputs a probability, so it's not immediately clear to me how $\mathcal{T}$ is actually defined.
* Line 339--340: More details should be given on how and why MARL beats droop control methods.

Minor:
* Line 193: "radius" --> "radians"
* Lines 194 and 195: "measures" --> "measurements"
* Line 218: Since "loss" can have multiple meanings in this context, it may be worth clarify that "loss" is used in the ML context here.

**Time Spent Reviewing:**

2

---

> ### Author Response · Authors · 2021-08-10
> **Response to reviewer S8R6**
>
> ### We thank the reviewer for insightful comments
>
> 1. > Key assumption: One of my main concerns is regarding the assumption made in the paper that multiple inverters are controlled by one company that has full information within its operational region. I do not believe this model is common in present distribution networks -- that is, where smart inverter control exists, it is generally the case that each inverter operates separately based on local information. The authors should provide much more context around this assumption (e.g., how does this grid operation mode relate to existing grid operation modes?), both to avoid inadvertently misleading the ML community about the state of current grid operations, and to better justify the technical assumptions on which the rest of the work is based.
>
>    ***Response:*** The authors agree with the reviewer that more context should be provided and clarified. The authors also agree that the common power system in practice is either locally or centrally controlled which are both considered as baselines in the paper. To tackle the increasing implementations of distributed energy resources (DERs), e.g. PVs, some existing works are studying on decentralized control option where PVs in each region are controlled by regional operator [6] (that is also in assumptions). To further give flexibility in future market-oriented power network, we investigate a new control mode where PVs in a region can be controlled by different operators. Therefore, customers can optimize their profits by comparing different operators. Nevertheless, these operators should follow the grid code and offer auxiliary survices such as voltage control. To do so, regional observations are shared. More details for the assumption in this paper can be referred to the response to question 3 for Reviewer DvqK. The authors will clarify this in the camera-ready version to give the ML community a clear understanding.
>
> 2. > Concerns on usage instructions.
>
>    ***Response:*** First, the interface of this environment is currently referring to Pymarl (i.e. a popular multi-agent framework from Whiteson Lab), so it is easy for freshers to start up. We plan to write a document to introduce this envrionment in details, including how to build up new networks and plug in new data. Since the backgrounds of the authors ranges from electric power systems to MARL. Therefore, the authors believe that the understanding and usage of this environment should be friendly in practice.
>
> 3. > Terminology and explanations: As one of the goals of the paper is to make the voltage control setting accessible to ML practitioners, the authors should be careful to go through and make sure all power systems terminology and assumptions are properly defined, e.g.,:
> Line 92: Define "power curtailment"
> Line 127--129: It would be important to clarify why and when the distribution network can be modeled as a tree graph (especially since the 33-bus system evaluated is actually not a tree, though the authors mention in the appendix that they modify it to remove loops).
> Line 140: Define "per unit"
>
>    ***Response:*** we now explain the terminology one by one.
> Line 92 "power curtailment" means restricting the usage of the maximum admissible PV active power.
> Line 127--129: The distribution network can have loop, e.g. interconnection line between different control areas, to improve the reliability and capacity of the power distribution system. However, in this paper we consider the distribution networks as tree (radial networks) that are the common setup in the electric power community. We note this setting is beneficial and specific for voltage control problem as the terminal users can directly output their extra active power toward the main grid, causing reverse power flow problem [2-4].
> Line 140: "per unit" means the power (or voltage) compared with a reference power (or voltage) that is referred as a unit. In power system, the voltage level and power level can be very high, e.g. ~100kv and 10MW which is not convenient for large scale computation [chapter 2, 1].
>
> 4. > Lines 41--42: Voltage and power are indeed related, through the power flow equations, which means that the constraint and control action are indeed linked. Would it be possible to clarify what is meant by this statement?
>
>    ***Response:*** In the paper, the authors indicated that constraint here means voltage and control action here means reactive power. From Eq. (1) it can be seen that the voltage (v) and the reactive power (q) should satisfy the balance equations called power flow equations that means they are coupled (linked).
>
> 5. > Line 140: "When the load is heavy during the nighttime" -- Why is load heavier at night, especially during seasons with no HVAC? Or is it just meant that net load is heavy due to low availability of PV power at night?
>
>    ***Response:***   HVAC represents the AC high-voltage tranmission lines on power community which is irrelevant to the question. During the night, the domestic electricity usage is increasing due to the high demand and the sharply electrcity demand variation might decrease the voltage level at the end user [5]. Notice that we not only consider the voltage rising caused by large PV penetration but simultaneously resolve the voltage decreasing casued by large power demand and variation.
>
> 6. > Line 184--186: The current notation for implies that it is a set of sets, rather than a union of sets.
>
>    ***Response:*** Yes. Thanks for the catch. It could be more appropriate to be changed to collection.
>
> 7. > Line 191: What is meant by "at the last step"?
>
>    ***Response:*** The last step here means the previous time step. We will change it to remove the ambiguity.
>
> 8. > Line 205, "State Transition Probability Function": The notation here should be cleaned up, as outputs a state but outputs a probability, so it's not immediately clear to me how is actually defined.
>
>    ***Response:*** In practice and implementation of simulations, $\delta(\mathbf{s}\_t, \mathbf{a}) \mapsto \mathbf{s}\_{t+\tau}$ and $Pr(\mathbf{s}\_{t+1} | \mathbf{s}\_{t+\tau})$ consecutively happen, where $\tau << \Delta t$ is the short interval less than the interval between two controls (i.e. a time step) and $\Delta t = 1$ is the interval between two controls considered in this paper. The authors think $\mathcal{T}(\mathbf{s}\_{t+1} | \mathbf{s}\_{t}, \mathbf{a}\_{t}) = Pr(\mathbf{s}\_{t+1} | \delta(\mathbf{s}\_{t}, \mathbf{a}\_{t}))$ could be more reasonable and clearer in mathmetical formulations. Thanks the reviewer for the good catch. The reviewer will change it in the camera-ready version.
>
> 9. > Line 339--340: More details should be given on how and why MARL beats droop control methods.
>
>    ***Response:*** In general, MARL can address the limitations described on Line 334-337. (1) Droop control can only generate reactive power to regulate the voltage of the bus with PV according to the local measurement, so that the other buses may not be under controlled. However, MARL is able to learn to generate reactive powers to regulate the voltage of all buses. (2) Droop control needs frequent interaction with the envrionment (to obtain measurements) to guarantee the control performance, while MARL actually intrinsically predicts the next state and conduct controls so that the frequency is less sensitive. (3) Droop control is sensitive to the frequency and the shape of q-v curve (see Appendix A.4) case by case, while most of MARL algorithms are only needed to tune some hyperparameters and can adaptively learn a policy that can fit general cases.
>
> ***Reference***
>
> [1] Gómez-Expósito, A., Conejo, A. J., & Cañizares, C. (Eds.). (2018). Electric energy systems: analysis and operation. CRC press.
>
> [2] Liu, Haotian, and Wenchuan Wu. "Online multi-agent reinforcement learning for decentralized inverter-based volt-var control." IEEE Transactions on Smart Grid (2021).
>
> [3] Xu, Y., Dong, Z. Y., Zhang, R., & Hill, D. J. (2017). Multi-timescale coordinated voltage/var control of high renewable-penetrated distribution systems. IEEE Transactions on Power Systems, 32(6), 4398-4408.
>
> [4] Tang, Z., Hill, D. J., & Liu, T. (2020). Distributed coordinated reactive power control for voltage regulation in distribution networks. IEEE Transactions on Smart Grid, 12(1), 312-323.
>
> [5] Varma, R. K., Khadkikar, V., & Seethapathy, R. (2009). Nighttime application of PV solar farm as STATCOM to regulate grid voltage. IEEE transactions on energy conversion, 24(4), 983-985.
>
> [6] Cao, D., Zhao, J., Hu, W., Ding, F., Huang, Q., Chen, Z., & Blaabjerg, F. (2021). Data-Driven Multi-agent Deep Reinforcement Learning for Distribution System Decentralized Voltage Control with High Penetration of PVs. IEEE Transactions on Smart Grid.

---

> > ### Comment · Reviewer_S8R6 · 2021-08-24
> > **Additional points of clarification**
> >
> > Thank you to the authors for the detailed response. Most of my points have been addressed. Some minor residual points:
> >
> > > Response: In the paper, the authors indicated that constraint here means voltage and control action here means reactive power. From Eq. (1) it can be seen that the voltage (v) and the reactive power (q) should satisfy the balance equations called power flow equations that means they are coupled (linked).
> >
> > Correct, I agree that these variables are coupled. However, the paper states that "there are no explicit relationship between the constraint (voltage) and the control action (reactive power)."
> >
> > > Response: HVAC represents the AC high-voltage tranmission lines on power community which is irrelevant to the question. During the night, the domestic electricity usage is increasing due to the high demand and the sharply electrcity demand variation might decrease the voltage level at the end user [5]. Notice that we not only consider the voltage rising caused by large PV penetration but simultaneously resolve the voltage decreasing casued by large power demand and variation.
> >
> > My apologies for using an overloaded acronym. By HVAC, I meant "heating, ventilation, and air conditioning" -- that is, I was asking if heating and cooling loads were increasing at night. Given this, could the authors please clarify? E.g. in the US in the summer, residential and commercial demand peaks in the middle of the day (due to air conditioning load), not overnight.
> >
> > > Response: In general, MARL can address the limitations described on Line 334-337. (1) Droop control can only generate reactive power to regulate the voltage of the bus with PV according to the local measurement, so that the other buses may not be under controlled. However, MARL is able to learn to generate reactive powers to regulate the voltage of all buses. (2) Droop control needs frequent interaction with the envrionment (to obtain measurements) to guarantee the control performance, while MARL actually intrinsically predicts the next state and conduct controls so that the frequency is less sensitive. (3) Droop control is sensitive to the frequency and the shape of q-v curve (see Appendix A.4) case by case, while most of MARL algorithms are only needed to tune some hyperparameters and can adaptively learn a policy that can fit general cases.
> >
> > I think the authors should be a bit more precise about how they address this. For instance, since the underlying system model is unlikely to be perfect, MARL too is likely to be dependent on system measurements. Also, note that neither droop nor MARL "guarantee" control performance, so the authors should be careful about the usage of this word.

---

> > > ### Author Response · Authors · 2021-08-26
> > > **Re: Additional points of clarification**
> > >
> > > 1. > Correct, I agree that these variables are coupled. However, the paper states that "there are no explicit relationship between the constraint (voltage) and the control action (reactive power)."
> > >
> > > **Response:** Although voltage and reactive power are coupled, there is an another condition over voltage (e.g., $\underline{v} \leq v \leq \overline{v}$) in addition to the power flow equations. In the paper, we mean this condition has no explicit relationship to reactive power, which is **not** in conflict with "these variables are coupled". We will clarify this in a clearer manner in the revised version.
> > >
> > > 2. > My apologies for using an overloaded acronym. By HVAC, I meant "heating, ventilation, and air conditioning" -- that is, I was asking if heating and cooling loads were increasing at night. Given this, could the authors please clarify? E.g. in the US in the summer, residential and commercial demand peaks in the middle of the day (due to air conditioning load), not overnight.
> > >
> > > **Response:** We are sorry to use imcomplete description on the statement 'When the load is heavy during the nighttime' in line 140. And we agree during the summer, the highest electricity consumption can be at mid of the day and decreases, though still high, during the night. We mainly follow the intuition from Fig.11 in [1] and Fig.13 in [2] where the electricity demand is highest at night, while ignoring there might be different load profiles in different regions and seasons. To answer the original question, the author wants to propose a situation when the power is extensively consumed by the end users so that a voltage decreasing problem might occur if there does not have control or planning. For example, in [2] Fig. 15, with $P_{pv}=0 $ at night, Fig. 15(a) shows that the voltage can be lower than 0.96 p.u. without var control and Fig. 15(b) verifies that the control option when the voltage is low is to provide reactive power by the PV inverter. We agree that this is related to $P_{pv}$, but we won't say this is because 'low availability of PV power at night' as the voltage drop exists regardless of the existence of the PV panels. However, by fully using the PV capacity, the MARL algorithm gives a new fast-response option to this problem without any further investment and control, such as tap transformer and capacitor bank.
> > >
> > > [1] Agalgaonkar, Y. P., Pal, B. C., & Jabr, R. A. (2013). Distribution voltage control considering the impact of PV generation on tap changers and autonomous regulators. IEEE Transactions on Power Systems, 29(1), 182-192.
> > > [2] Singhal, A., Ajjarapu, V., Fuller, J., & Hansen, J. (2018). Real-time local volt/var control under external disturbances with high PV penetration. IEEE Transactions on Smart Grid, 10(4), 3849-3859.
> > >
> > > 3. > I think the authors should be a bit more precise about how they address this. For instance, since the underlying system model is unlikely to be perfect, MARL too is likely to be dependent on system measurements. Also, note that neither droop nor MARL "guarantee" control performance, so the authors should be careful about the usage of this word.
> > >
> > > **Response:** We agree with the comment on the usage of "guarantee". Actually, the "guarantee" we mean in the last response is "maintain". We will be careful on this in the writing.
> > >
> > > Next, we would give more clarifications on how MARL can address these problems of droop controls. (1) Since droop control can only regulate the voltage on the buses with PVs by local measurement, MARL is able to address it via learning coordination with each other, given the global reward with the whole picture of buses. (2) Since droop control needs to find an optimal solution with interaction to environment multiple times given a state **that is difficult to fit the high frequent changes of loads and PVs**, MARL can address it by learning a policy that fits the dynamics and directly gives the decision for a state (fast response). (3) Since droop control is relied on the hyperparameters (i.e., the shape of q-v curve) that needs to be manually tuned to fit **the specific scenarios of power networks**, MARL can address it through learning policy by tuning hyperparameters that can fit most of scenarios (a type of genenralisation).
> > >
> > > We wish our answers can address your concerns. If there is other questions, we are pleased to discuss with you. Thanks!

---

### Official Review · Reviewer_YRiZ · 2021-07-25

**Rating:** 6
**Confidence:** 3

**Summary:**

In this work, the authors studied applying multi-agent reinforcement learning (MARL) to active voltage control in the power distribution networks, and targeting for bridging the gap between the power community and MARL community. Main contributions include:
1) develop an open-source environment of active voltage control for MARL and formally define this problem as a
65 Dec-POMDP;
2) conduct large scale experimentation with 8 state-of-the-art MARL algorithms on different scenarios of voltage control problem;
3) found valuable observations, e.g.,  notice the importance of converting voltage constraints to auxiliary reward functions and verify the sensitivity of MARL performances to the auxiliary reward functions, with a new bowl-shape auxiliary reward proposed which stands out in the test results;
4) by analyzing the experimental results, the authors imply the possible difficulties of applying MARL and the further research directions. The authors also conducted the comparison with traditional control methods, and empirically show the rationality and hope to use MARL as a solution for active voltage control.

**Limitations And Societal Impact:**

Yes

**Main Review:**

This paper is clearly written and well-organized. The main objective is to present/formulate a problem in power networks as an exciting and yet challenging real-world scenario for application of multi-agent reinforcement learning (MARL), developing an open-source environment of active voltage control for MARL, conducting some initial empirical study, and summarizing the main observations (including some key observed challenges/limitations of existing MARL techniques in such problem and some potential future research directions). It mainly targets for bridging the gap between the power community and MARL community in this research topic.

I think the problem is interesting and the paper contains valuable results. Bridging the gap between MARL community and power community is important, and the research work here might potentially motivate more future follow-up research in this direction and would also be helpful for real world large scale application of MARL in power networks.

However, for publication of this result on a top MARL/ML conference like NIPS, I'm still not fully convinced by the novelty and contribution of this study. The main contribution of this this paper is more about formulating the active voltage control problem in power network as MARL problem, conducted some initial study of existing MARL techniques in such problem, and identified some key limitations/difficulties there. If the authors could have some results where they addressed some of these identified limitations and made some improvements (which can be somewhat customized to such power network domain) to the existing MARL techniques, then I'd be more convinced that its contribution is sufficient for publication as a regular paper on such a top tier MARL/ML conference like NIPS.

On the other hand, to be clear, I do think this paper indeed studies an interesting problem and it contains valuable research results that might be good for publication at some good conference/journals, and it will be beneficial for future research, especially for interaction between power community and MARL community. But I just think the contribution and novelty in current manuscript might not be strong enough for publication as a regular paper on such a top tier MARL/ML conference like NIPS.

Update: I've read all the review discussions here and checked out all the detailed responses and clarifications from the authors. I'm slightly more convinced about the contribution and novelty of this paper, and I decided to upgrade my rating from 5 to 6. Though I do agree with several other reviewers that, even after the responses and clarifications from the authors, this paper still seems to be quite at the border line.

**Time Spent Reviewing:**

2.5 hr

---

> ### Author Response · Authors · 2021-08-10
> **Response to reviewer YRiZ**
>
> ### The authors would thank the reviewer for insightful and fair comments.
>
> As the reviewer noticed, this paper mainly aims to bridge the gap between the MARL community and the electric power community. The authors agree that if there is an improvement (or change) based on the existing MARL algorithms adaptive to this problem, then this paper is stronger to be accepted. Nevertheless, we would like to argue that the existing works in this paper deserve to be published on NeurIPS.
>
> First, the definition of the problem and the understanding of the feasibility of MARL to the problem are also important compared with giving a novel method. Many papers that claim to solve real-world application problems usually ignore the description and the analysis of the problem itself for leaving space to introduce the novel method. This is not friendly to the development of MARL application to real-world problems. For this reason, we focus on introducing the new problem itself and attempt to give preliminary experimental results and analysis with the existing state-of-the-art MARL algorithms so that more researchers from the MARL community can have a clear understanding on this problem and contribute to it.
>
> As for the novelty in this paper, most of the related works in the power community focus on the scenario with a region as a control unit (i.e., decentralised), where the main purpose is to reduce the number of agents so that it is easy to implement, e.g., only 4-5 agents in a network with 323 buses [1]. However, this is not suitable for operations over renewable energy equipments (e.g. PV inverters) in the future (see the response to question 3 for reviewer DvqK for details), and this is the reason why we propose the distributed active voltage control problem, which is a novel and meaningful setup that was never attempted before. Meanwhile, the detailed analysis and comparison among so many state-of-the-art MARL algorithms was never accomplished in the power community.
>
> Finally, in future work we plan to design new algorithms that can solve the limitations mentioned in this paper as the reviewer said. We wish the reviewer can consider the reasons we stated and make a new decision.
>
> ***Reference***
>
> [1] Cao, D., Zhao, J., Hu, W., Ding, F., Huang, Q., Chen, Z., & Blaabjerg, F. (2021). Data-Driven Multi-agent Deep Reinforcement Learning for Distribution System Decentralized Voltage Control with High Penetration of PVs. IEEE Transactions on Smart Grid.

---

### Author Response · Authors · 2021-08-11
**Thanks to all Reviewers**

Dear all reviewers,


The authors would thank all reviewers for the efforts on reviewing this paper and raising thoughtful and insighful questions.

The authors have tried the best to address the concerns from all reviewers and hope reviewers can be satisfied after reading the reponses.

If reviewers have any other questions, the authors are very happy to continue discussions! Thanks again to all reviewers' efforts.


Best,

The authors of Paper 2855

---

### Decision · Program_Chairs · 2021-09-27

**Decision:**

Accept (Poster)

**Comment:**

This paper presents an multi-agent RL environment for active voltage control on power distribution networks. This could be a new interesting and practicality-oriented benchmark environment for multi-agent RL research. Most of the reviewers' concerns were (1) lack of access to the implementation of the simulator, which authors explicitly promised to make the code public if the paper gets accepted, and (2) lack of reference to another power network environment L2RPN / GridOp, which is not exactly active voltage control but should have been referenced. AC is recommending borderline accept, and if the paper gets accepted, please address (1) and provide detailed comparison to L2RPN / GridOp in the related work section, so that readers understand what exact power network problem the paper is tackling.